# CELL: A Causal Perspective for Fairness-aware Graph Adaptation

**Hourun Li**[1]  **Yifan Wang**[2*]  **Qinghua Ran**[3]  **Junyu Luo**[3]  **Jia Yang**[3]  **Changling Zhou**[3]
**Zhiping Xiao**[4]  **Wei Ju**[3]  **Xiao Luo**[5*]  **Ming Zhang**[1*]

## Abstract

This paper studies fairness-aware graph adaptation, aiming to transfer knowledge from a labeled source graph to an unlabeled target graph while addressing fairness. Most prior methods require target-domain attributes to learn invariant graph representations of sensitive attributes, which are often unavailable in practice. To address this limitation, we introduce Causality-attended Representation Dientanglement with Structural Alignment (CELL) for fairness-aware graph adaptation without requiring target sensitive labels. CELL constructs a causal graph to model the graph-generation mechanism and guide fair representation disentanglement. Specifically, CELL uses sensitive and causal encoders to extract sensitive and causal factors, respectively, and promotes disentanglement by minimizing their conditional mutual information. To leverage unlabeled target data, we further generate pseudo-labels for both target task labels and sensitive attributes, and use similarity relations to derive unbiased node representations. Finally, to further mitigate domain shift, we build a fairness-aware bipartite graph that provides additional structural supervision for cross-domain alignment. Experiments on benchmarks show that CELL consistently outperforms strong baselines in both predictive performance and fairness.

---

[1]State Key Laboratory for Multimedia Information Processing, School of Computer Science, PKU-Anker LLM Lab, Beijing Key Laboratory of Software and Hardware Cooperative Artificial Intelligence Systems, Peking University, Beijing, China [2]School of Artificial Intelligence and Data Science, University of International Business and Economics, Beijing, China [3]Peking University, Beijing, China [4]Paul G. Allen School of Computer Science and Engineering, University of Washington, Seattle, WA, USA [5]Department of Statistics, University of Wisconsin-Madison, USA. Correspondence to: Yifan Wang <yifanwang@uibe.edu.cn>, Xiao Luo <xiao.luo@wisc.edu>, Ming Zhang <mzhang_cs@pku.edu.cn>.

*Proceedings of the 43$^{rd}$ International Conference on Machine Learning*, Seoul, South Korea. PMLR 306, 2026. Copyright 2026 by the author(s).

## 1. Introduction

Graph Neural Networks (GNNs) have become the *de facto* approach for modeling complex relational structures, broadening their social impact (Kipf & Welling, 2017; Xu et al., 2019). By capturing node features and relational dependencies via message passing, GNNs have achieved remarkable success in various graph-based tasks (Wu et al., 2020; Ju et al., 2024). Node classification, which predicts node labels, is among the most fundamental tasks in graph learning and underpins applications such as community detection (Bianchi et al., 2020), cross-modal retrieval (Qian et al., 2022), and molecular property prediction (Wang et al., 2021). However, graph learning suffers from fairness issues caused by inherent graph-data biases, which is further amplified by GNN message passing (Chen et al., 2024).

To mitigate this issue, recent studies have explored fair graph learning, which is commonly divided into pre-, in-, and post-processing approaches according to when fairness interventions are applied. Pre-processing methods reduce bias before training by modifying the input graph, such as masking node features (Köse & Shen, 2021) or rewiring structures (Spinelli et al., 2021; Dong et al., 2022). In-processing methods impose fairness constraints during training via regularization (Agarwal et al., 2021; Jiang et al., 2024; Wang et al., 2025; Zhang et al., 2026), adversarial debiasing (Ling et al., 2023), or disentangled representation learning (Zhu et al., 2024a; Lee et al., 2025). Post-processing methods instead adjust trained model outputs to reduce bias and improve fairness (Dai & Wang, 2021; Zhang et al., 2024b).

However, most prior studies require sensitive attributes, such as race and gender, to be observed for all training nodes, which is often unrealistic in real-world scenarios because of privacy constraints or the absence of complete demographic information. Graph transfer learning offers a potential way to overcome this limitation by transferring useful knowledge from an attribute-rich source graph to a target graph with limited or unavailable labels (Han et al., 2021; Zhu et al., 2021; 2024b). Under this setting, graph domain adaptation (GDA) has been widely explored as a key technique, where the source and target domains are aligned within the representation space learned by GNNs. This line of research has attracted increasing attention in recent years (Qiao et al.,

2023; You et al., 2023; Liu et al., 2024; Fang et al., 2026; 2025). Accordingly, an important question arises: *When sensitive labels are missing in the target domain, how can fairness knowledge be transferred across graphs in an effective manner?* Driven by this motivation, we define a new problem setting, namely graph fairness adaptation.

Nonetheless, graph fairness adaptation for node classification remains challenging in two aspects. ❶ *Sensitive Group Distribution Shift under Domain Discrepancy.* Structural and semantic shifts between source and target graphs can change sensitive-group distributions, hindering fairness knowledge transfer and weakening target-domain bias mitigation. Ding et al. (Ding et al., 2021) showed that the fairness of an income prediction model optimized or evaluated in one state may not generalize to another. ❷ *Alignment Collapse during Entangled Information Transfer.* Existing alignment methods based on distribution discrepancy minimization (Wu et al., 2023) or adversarial learning (Dai et al., 2022) may entangle target-relevant and sensitive-related information, causing conflicts between fairness and performance and potentially leading to alignment collapse.

Towards this end, we propose a framework named Causality-attended Representation Dientanglement with Structural Alignment (CELL) for graph fairness adaptation, which aims to transfer the fairness knowledge from the source graph to the unlabeled target graph. Specifically, our CELL incorporates a dual graph encoder with a two-fold mutual information (MI) constraint, enabling the model to disentangle task-relevant and sensitive-related representations. Based on this, we generate pseudo-sensitive labels for target graph nodes and partition them into corresponding demographic groups. To enhance fairness in pseudo-labeling, group-acquired unbiased learning strategy explicitly emphasizes negative pairs sharing identical sensitive labels. Finally, for each target graph node, we retrieve source nodes with the same target while distinct sensitive labels for bipartite graph construction between two domains and employ a bipartite-aware domain alignment to decorrelate sensitive and target information for fairness adaptation.

In a nutshell, the contributions of CELL are as follows: ❶ *New Perspective.* We highlight the limited or unavailable nature of sensitive information in graph fairness learning and introduce an underexplored yet graph fairness adaptation problem. As far as we know, this study is the first attempt to explore this problem. ❷ *Novel Methodology.* We propose a novel framework termed CELL, which generalizes both target and sensitive representations with a dual graph encoder under a two-fold MI constraint and performs group-acquired bipartite alignment for unbiased domain alignment. ❸ *Extensive Experiments.* To evaluate CELL, we carry out extensive experiments on multiple benchmark datasets. The experimental findings indicate that our framework consistently achieves superior adaptation performance and enhanced fairness. The code is available at `https://github.com/HourunLi/CELL_ICML_2026.git`.

## 2. Preliminaries & Problem Definition

**Notations.** Let the *source domain graph* be denoted as $\mathcal{G}^{so} = \{\mathcal{V}^{so}, \mathcal{E}^{so}, \boldsymbol{X}^{so}, \boldsymbol{Y}^{so}, \boldsymbol{S}^{so}\}$, where $\mathcal{V}^{so}$ and $\mathcal{E}^{so}$ represent the node and edge set respectively. We use the adjacency matrix $\boldsymbol{A}^{so}$ to describe the structure information of the source domain graph, where $\boldsymbol{A}^{so}_{uv} = 1$ if there is an edge $(u, v) \in \mathcal{E}^{so}$, otherwise $\boldsymbol{A}^{so}_{uv} = 0$. The node feature matrix is given by $\boldsymbol{X}^{so} \in \mathbb{R}^{|\mathcal{V}^{so}| \times d}$, where each row $\boldsymbol{x}_v \in \mathbb{R}^d$ represents the $d$-dimensional feature vector of node $v$. The node sensitive attributes are specified as $\boldsymbol{S}^{so} = \{s_1, \ldots, s_{|\mathcal{V}^{so}|}\} \in \{0, 1\}^{|\mathcal{V}^{so}|}$, where $s_v$ is the sensitive label of node $v$. The binary node classification and the source node label matrix can be $\boldsymbol{Y}^{so} = \{y_1, \ldots, y_{|\mathcal{V}_{so}|}\} \in \{0, 1\}^{|\mathcal{V}^{so}|}$. Similarly, the *target domain graph* is denoted as $\mathcal{G}^{ta} = \{\mathcal{V}^{ta}, \mathcal{E}^{ta}, \boldsymbol{X}^{ta}\}$ with completely unlabeled node set $\mathcal{V}^{ta}$ and edge set $\mathcal{E}^{ta}$. Note that to facilitate alignment, we construct a unified feature space across the source and target domain graphs.

**Definition 2.1.** The sensitive group of the source and target domain graph is partitioned by nodes according to their sensitive attribute, formally defined as:

$$\mathcal{V}_s^* = \{v \in \mathcal{V}^* | s_v = s\}, * \in \{so, ta\}. \tag{1}$$

**Definition 2.2.** The Equalized Odds (EO) group of the graph is formed by partitioning nodes according to both target label $y$ and sensitive attribute $s$ of the node:

$$\mathcal{V}_{y,s}^* = \{v \in \mathcal{V}^* | (s_v = s) \cap (y_v = y)\} * \in \{so, ta\}. \tag{2}$$

**Definition 2.3.** Demographic parity (Calders et al., 2009) is achieved to ensure fairness by enforcing that nodes from different demographic groups have equal probabilities of being assigned positive predictions. Accordingly, $\Delta_{DP}$ of target domain graph can be:

$$\Delta_{DP} = |\mathbb{E}_{u \in \mathcal{V}^{ta}}(\hat{y}_u = 1 | s_u = 1) - \mathbb{E}_{v \in \mathcal{V}^{ta}}(\hat{y}_v = 1 | s_v = 0)|, \tag{3}$$

where $\hat{y}_v$ and $y_v$ represent the predicted and ground-truth label of the crossponding node. The dependence between predictions $\hat{y}$ and sensitive attribute $s$, namely $\hat{y} \perp\!\!\!\perp s$, is measured by $\Delta DP$.

**Definition 2.4.** Equalized odds (Hardt et al., 2016) stipulates fairness by ensuring that the True Positive Rate (TPR) and False Positive Rate (FPR) are identical across demographic groups. Formally, $\Delta_{EO}$ of target domain graph is defined as:

$$\Delta_{EO} = \frac{1}{2} \sum_{y=0}^{1} |\mathbb{E}_{u \in \mathcal{V}^{ta}}(\hat{y}_u = y | y_u = y, s_u = 1) \\ - \mathbb{E}_{v \in \mathcal{V}^{ta}}(\hat{y}_v = y | y_v = y, s_v = 0)|. \tag{4}$$

Note that $\Delta_{EO}$ quantifies the conditional independence between the predicted label $\hat{y}$ and sensitive attribute $s$ given the ground-truth label $y$, i.e., $\hat{y} \perp\!\!\!\perp s \mid y$.

**Problem Definition.** Graph fairness adaptation aims to leverage the target-label and sensitive-attribute information from a labeled source-domain graph and transfer it to a fully unlabeled target-domain graph. Formally, let $\mathcal{G}^{so}$ denote the labeled source-domain graph and $\mathcal{G}^{ta}$ denote the unlabeled target-domain graph. Under the covariate shift assumption (Ben-David et al., 2006; 2010), namely $\mathbb{P}(\mathcal{G}_{so}) \neq \mathbb{P}(\mathcal{G}_{ta})$ and $\mathbb{P}(\boldsymbol{Y}|\mathcal{G}_{so}) = \mathbb{P}(\boldsymbol{Y}|\mathcal{G}_{ta})$, the task is to predict node labels in $\mathcal{G}^{ta}$ while achieving both competitive performance and fairness.

# 3. The Proposed CELL

## 3.1. Framework Overview

This section presents CELL, which enables unbiased adaptation through group-acquired bipartite alignment by disentangling the transfer of target-relevant features from that of sensitive features. Figure 1 illustrates the overall architecture of our framework, while the following subsections provide detailed explanations of its individual components.

## 3.2. Causality-attended Representation Disentanglement with Mutual Information Optimization

**Causal Graph Construction.** Since the key challenge of the graph fairness adaptation lies in identifying stable sensitive-free features that preserve fair prediction while suppressing sensitive-aware features across domains, we perform feature disentanglement based on the constructed causal graph. We formalize the dependencies between variables through a Structural Causal Model (SCM) (Pearl et al., 2016; Luo et al., 2025; Sui et al., 2022), where the three key mechanisms can be defined as:

- Domain Latent Factorization: $C^* \leftarrow \mathcal{D}^*, * \in \{so, ta\} \rightarrow S^*$ ensures that task factor $C^*$ is preserved, while permitting residual dependencies attributed to sensitive factors $S^*$ across the source domain $\mathcal{D}^{so}$ and target domain $\mathcal{D}^{ta}$.

- Graph Generation: $C^* \rightarrow \mathcal{G}^* \leftarrow S^*$ specifies that the observed graph data of two domains $\mathcal{G}^*$ is generated through the causal variable and the bias fairness-aware variable.

- Label Determination: $C^* \rightarrow Y^*$ indicates that the causal variable is the only endogenous parent to determine the ground-truth task label $Y^*$ under distribution shift.

Note that the spurious correlations of $C^*$ and $S^*$ within and between the graphs lead to poor fairness generalization under distribution shifts.

**Feature Disentanglement for Fairness Preservation.** Following the causal theory (Wu et al., 2022), there exist a directed link between the variable $Y$ to its parent $PA(Y)$ in an SCM, if and only if a causal mechanism $Y = H(PA(Y), \epsilon_Y)$ persists, where the $\epsilon_Y \perp\!\!\!\perp PA(Y)$ is the exogenous noise of $Y$. In our setting, the mechanism can be represented as:

$$Y = H(PA(Y), S), Y \perp\!\!\!\perp S|C. \tag{5}$$

Thus, we disentangle $S$ and $C$ to preserve the causal effect of $C$ on $Y$, while eliminating the influence of $S$ for graph fairness adaptation. Specifically, we employ GNNs to obtain the node embeddings $\boldsymbol{Z}_s^{so}$ of the source domain graph and incorporate a sensitive discriminator $\xi(\cdot) : \boldsymbol{Z}_s^{so} \rightarrow \boldsymbol{S}$ to explicitly correlate with the sensitive attribute. The classification loss can be defined as:

$$\mathcal{L}_{cls}^s = - \sum_{v \in \mathcal{V}^{so}} \text{BCE}(s_v, \xi(\boldsymbol{z}_{v,s}^{so})). \tag{6}$$

Based on the sensitive-aware embeddings, we employ another GNN to generate the causal task-relevant embedding $\boldsymbol{Z}_c^{so}$ and implement the graph fairness learning through:

$$\max \underbrace{I(\boldsymbol{Z}_c^{so}; \mathcal{D}^{so}|\boldsymbol{Z}_s^{so})}_{\text{Conditional Fair Prediction}} - \beta \underbrace{I(\boldsymbol{Z}_s^{so}; \boldsymbol{Z}_c^{so})}_{\text{Sensitive-Free}}, \tag{7}$$

where $I(\cdot, \cdot)$ denotes the mutual information (Zhao et al., 2023; Liu et al., 2022). In practice, we derive a sample-based MI upper bound based on the Contrastive Log-ratio Upper Bound (CLUB) (Cheng et al., 2020) for the aforementioned sensitive-free constraints, which can be formulated as:

$$\min I(\boldsymbol{Z}_s^{so}; \boldsymbol{Z}_c^{so}) := \mathcal{L}_{MI}^s$$
$$= \frac{1}{M} \sum_{v=1}^{M} \left[ \log q_\theta(\boldsymbol{z}_{v,s}^{so}|\boldsymbol{z}_{v,c}^{so}) - \frac{1}{M} \sum_{u=1}^{M} q_\theta(\boldsymbol{z}_{u,s}^{so}|\boldsymbol{z}_{v,c}^{so}) \right], \tag{8}$$

where we leverage a Multilayer Perceptron (MLP) $q_\theta(\boldsymbol{z}_{v,s}^{so}|\boldsymbol{z}_{v,c}^{so})$ to approximate the conditional probability. For the requirement of the conditional fair prediction constraint, we leverage a model pre-trained with the target task label to generate the low-rank node embedding $\boldsymbol{\mu}_v$ given the high-dimension and sparsity of the source domain graph. Then, we employ a Conditional InfoNCE (Gupta et al., 2021) for the MI lower-bound, which can be defined as:

$$\max I(\boldsymbol{Z}_c^{so}; \mathcal{D}^{so}|\boldsymbol{Z}_s^{so}) := \mathcal{L}_{MI}^c$$
$$= \mathbb{E} \left[ \log \frac{\exp f(\boldsymbol{\mu}_v, \boldsymbol{z}_{v,c}^{so}, \boldsymbol{z}_{v,s}^{so})}{\frac{1}{M} \sum_{u=1}^{M} \exp f(\boldsymbol{\mu}_u, \boldsymbol{z}_{v,c}^{so}, \boldsymbol{z}_{v,s}^{so})} \right], \tag{9}$$

where $\boldsymbol{\mu}_v \sim p(\boldsymbol{\mu}_v|\boldsymbol{z}_{v,s}^{so})$ and $f(\cdot)$ denotes the conditional score function, where we implement as the weighted cosine similarity (Zhao et al., 2023):

$$f(\boldsymbol{\mu}_v, \boldsymbol{z}_{v,c}^{so}, \boldsymbol{z}_{v,s}^{so}) = \text{sim}(\boldsymbol{\mu}_v, \boldsymbol{z}_{v,c}^{so} + \alpha \boldsymbol{z}_{v,s}^{so}), \tag{10}$$

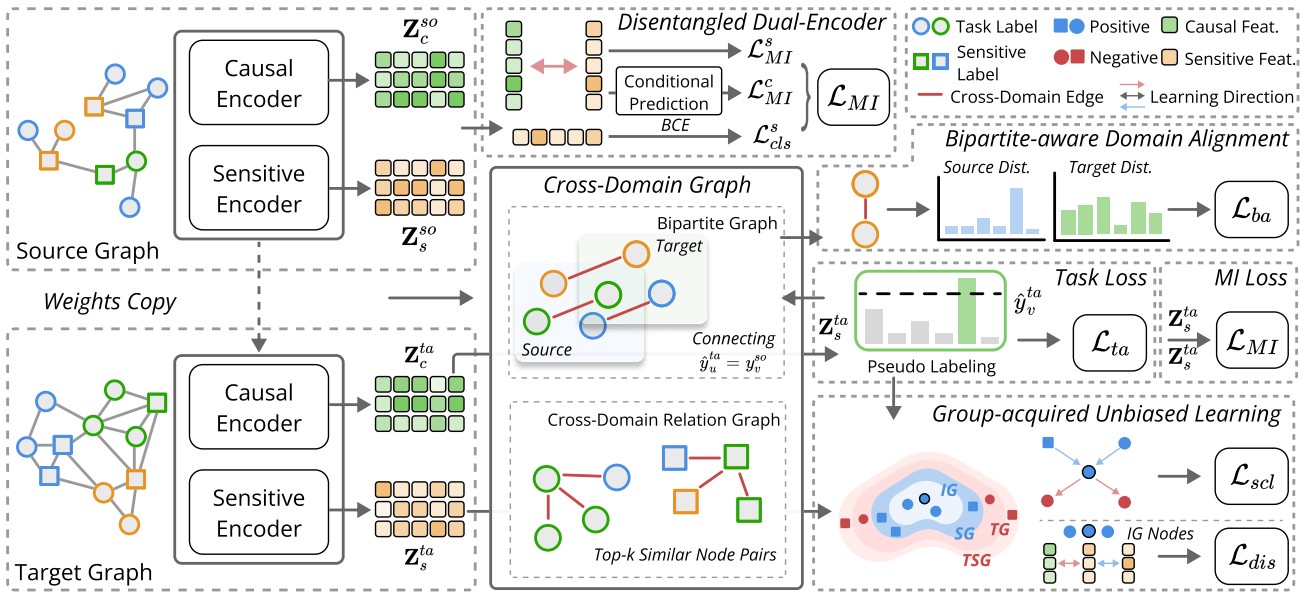

*Figure 1.* Illustration of CELL with three modules: (1) Causality-attended Disentanglement: MI-based encoders disentangle task-relevant and sensitive factors. (2) Group-attended Unbiased Learning: Generate pseudo labels to support group-attended unbiased learning (3) Bipartite Domain Alignment: Reduce domain shift through clustering-based bipartite graph alignment

where $\alpha$ is the trade-off hyper-parameter and $\text{sim}(\cdot)$ is the cosine similarity function. Since the sensitive attribute induces a bias direction, we implement the conditional sampling based on the direction function, namely $\{\boldsymbol{\mu}_{v'}|\pi(\boldsymbol{z}_{v,c}^{so}, \boldsymbol{z}_{v',c}^{so}) > 0\}$. So in this way, we can ensure that the sampled embedding belongs to the same bias direction w.r.t. the conditional distribution. The composite fairness learning objective for the source domain graph is $\mathcal{L}_{MI} = \mathcal{L}_{cls}^s + \mathcal{L}_{MI}^s - \mathcal{L}_{MI}^c$.

### 3.3. Group-attended Pseudo-labeling for Unbiased Representation Learning

To mitigate the label scarcity (Cho & Lee, 2025) of the target domain graph while preventing fairness in pseudo-labels, we develop a fairness-preserving discriminative learning mechanism with group-acquired enhancement. Specifically, we employ the dual-encoder to get the node embedding $\boldsymbol{Z}_c^{ta}$ and $\boldsymbol{Z}_s^{ta}$ of the target domain graph and leverage them to determine the corresponding pseudo-label distribution,

$$\hat{y}_v^{ta} = \arg\max \psi(\boldsymbol{z}_{v,c}^{ta}), \ \hat{s}_v^{ta} = \arg\max \xi(\boldsymbol{z}_{v,s}^{ta}), \quad (11)$$

where $\psi(\cdot)$ projects the causal feature to the task label space. To preserve fairness in pseudo-label predictions, we decompose node similarities in the target domain graph into fine-grained types w.r.t. demographic groups, and explicitly penalize the model when it captures spurious correlations or sensitive information (Park et al., 2022; Zhang et al., 2024a; Wang et al., 2025). Specifically, given an anchor node $v$, we categorize its similarity relations into three groups defined as follows.

- *Intra-Group* (**IG**): Similarity is computed with respect to the EO group, where the corresponding node group is given by $\mathcal{V}_{ig}^{ta}(v) = \{ig \in \mathcal{V}^{ta}|s_{ig} = \hat{s}_v^{ta} \cap y_{ig} = \hat{y}_v^{ta}\}$.

- *Sensitive Inter-Group* (**SG**): We define the similarity between an anchor and nodes that belong to the same target class but distinct sensitive attributes, with the corresponding node group formally denoted as $\mathcal{V}_{sg}^{ta}(v) = \{sg \in \mathcal{V}^{ta}|s_{sg} \neq \hat{s}_v^{ta} \cap y_{sg} = \hat{y}_v^{ta}\}$.

- *Target Inter-Group* (**TG**): The similarity is characterized by the relationship between an anchor node and those nodes sharing its sensitive attribute while belonging to different target classes. Accordingly, the node group is given by $\mathcal{V}_{tg}^{ta}(v) = \{tg \in \mathcal{V}^{ta}|s_{tg} = \hat{s}_v^{ta} \cap y_{tg} \neq \hat{y}_v^{ta}\}$.

Building upon this, we encourage higher similarity among nodes within *IG* and *SG* than among those within *TG* to promote fairness in the target domain graph, formulated as:

$$\mathcal{L}_{scl} = -\frac{1}{|\mathcal{V}^{ta}|}\sum_{v \in \mathcal{V}^{ta}}\frac{1}{|\mathcal{V}_c(v)|}\sum_{u \in \mathcal{V}_c(v)}\log\frac{\phi_{u,c}}{\sum_{tg \in \mathcal{V}_{tg}^{ta}(v)}\phi_{tg,c}}, \quad (12)$$

where $\mathcal{V}_c(v) = \mathcal{V}_{ig}^{ta}(v) \cup \mathcal{V}_{sg}^{ta}(v)$ and $\phi_{*,c} = \exp(\boldsymbol{z}_{v,c}^{ta} \cdot \boldsymbol{z}_{*,c}^{ta}/\tau), * \in \{u, tg\}$. Meanwhile, we further decorrelate the sensitive information within *IG* of the target domain graph, defined as:

$$\mathcal{L}_{dis} = -\frac{1}{|\mathcal{V}^{ta}|}\sum_{v \in \mathcal{V}^{ta}}\frac{1}{|\mathcal{V}_s(v)|}\sum_{u \in \mathcal{V}_{ig}^{ta}(v)}\log\frac{\phi_{u,s}}{\sum_{ig \in \mathcal{V}_{ig}^{ta}(v)}\phi_{ig,c}}, \quad (13)$$

where $\phi_{u,s} = \exp(\boldsymbol{z}_{v,c}^{ta} \cdot \boldsymbol{z}_{u,s}^{ta}/\tau)$. The group-acquired unbiased loss is $\mathcal{L}_{ub} = \mathcal{L}_{scl} + \mathcal{L}_{dis}$, which emphasizes negative pairs with the same sensitive labels to promote fairness under distribution shift. Comparing with the previous work (Wang et al., 2025), we utilize pseudo-labels to formalize demographic groups.

### 3.4. Domain Alignment with Fairness-aware Bipartite

Although the pseudo-labels are generated for the target domain graph, the severe shift between source and target domains persists, which may lead to unreliable supervision signals (Li et al., 2023). To mitigate this, we construct a bipartite graph where edges connect similar sensitive-free node pairs across domains and introduce a bipartite-aware mechanism for the domain alignment (Zhang et al., 2025; Chen et al., 2021). Specifically, given the node $u \in \mathcal{V}^{ta}$ from target domain graph with pseudo-labels $\hat{y}_u^{ta}$ and $\hat{s}_u^{ta}$, we retrieve nodes $v \in \mathcal{V}^{so}$ from source domain graph and add edges between pair $(u, v)$ as:

$$\boldsymbol{B}_{uv} = \begin{cases} 1, & \hat{y}_u^{ta} = y_v^{so} \cap \hat{s}_u^{ta} \neq s_v^{so} \\ 0, & \text{otherwise} \end{cases}, \quad (14)$$

where $\boldsymbol{B}$ denotes the adjacency matrix of the bipartite graph, with each source and target domain sample being a node. To facilitate domain alignment, we impose a graph clustering constraint on the label predictions, enforcing that the majority of connected edges lie within the same clusters, which not only reduces domain discrepancy but also strengthens the discriminative capacity of task-relevant embeddings. The bipartite alignment loss can be defined as:

$$\mathcal{L}_{ba} = \|(\boldsymbol{I} - \boldsymbol{L}) - \boldsymbol{P}\boldsymbol{P}^T\|_F^2, \quad (15)$$

where $\boldsymbol{P}$ is the label prediction matrix, which is constructed as $\boldsymbol{P} = [\mathbb{1}(\boldsymbol{Y}^{so}), \xi(\boldsymbol{Z}_c^{ta})]$ with $\mathbb{1}[\cdot]$ as the one-hot function. $\boldsymbol{L}$ denotes the normalized Laplacian matrix of $\boldsymbol{B}$, and $\boldsymbol{I}$ is the identity matrix. Note that we ignore the intra-domain relationships and the above loss can be rewritten as:

$$\begin{aligned} \mathcal{L}_{ba} = &-2 \sum_{u,v} \frac{\boldsymbol{B}_{uv}}{\sqrt{d_u}\sqrt{d_v}} \boldsymbol{p}_u^T \boldsymbol{p}_v \\ &+ \sum_{u,v} \boldsymbol{B}_{uv}(\boldsymbol{p}_u^T \boldsymbol{p}_v)^2 + \text{const}, \end{aligned} \quad (16)$$

where $d_u$ denotes the degree of node $u$ in the bipartite graph.

### 3.5. Overall Optimization

To alleviate label scarcity while avoiding overconfidence in target domain pseudo-labels, we also quantify the prediction certainty through maximum class probability:

$$m_v^{ta} = \max_{k'} \psi(\boldsymbol{z}_{v,c}^{ta})[k], \quad (17)$$

where $m_v^{ta}$ is the confidence score. Then, we introduce an adaptive confidence score $\tau_k$ for class $k$ based on the estimated prediction certainty, defined as

$$\begin{aligned} \tau_k = \mathcal{M}_k \cdot \tau, \ \mathcal{M}_k \\ = \max\{m_v^{ta} | \arg\max_{k'} \psi(\boldsymbol{z}_{v,c}^{ta})[k'] = k\} \end{aligned} \quad (18)$$

where $\tau$ denotes the threshold. And the confident set $\mathcal{C}$ of the target domain graph can be refined as:

$$\mathcal{C} = \{v | v \in \mathcal{V}^{ta}, k = \arg\max_{k'} \psi(\boldsymbol{z}_{v,c}^{ta})[k'], m_v^{ta} > \tau_k\}. \quad (19)$$

We further optimize the model within the confident set $\mathcal{C}$ for cross-domain stability:

$$\mathcal{L}_{ta} = -\frac{1}{|\mathcal{C}|} \sum_{v \in \mathcal{C}} \log \psi(\boldsymbol{z}_{v,c}^{ta})[\hat{y}_v^{ta}]. \quad (20)$$

Similarly, we also leverage a threshold $\delta$ to filter out nodes with a high-confidence sensitive label for group-acquired enhancement. The overall objective of our graph fairness adaptation framework is:

$$\mathcal{L} = \mathcal{L}_{MI} + \beta\mathcal{L}_{ub} + \gamma\mathcal{L}_{ba} + \eta\mathcal{L}_{ta} \quad (21)$$

where $\beta$, $\gamma$ and $\eta$ denote the hyperparameter to balance each component.

### 3.6. Theoretical Analysis

**Theorem 3.1.** *(Fairness Upper Bound)*

*Let classifier $h$ depend only on $\boldsymbol{Z}_c$. Assume $h$ is L-lipschitz in distribution shift. Define*

$$\begin{aligned} \Delta_{EO} = \sum_{y \in \{0,1\}} |P(h = 1|S = 0, Y = y) - \\ P(h = 1|S = 1, Y = y)|. \end{aligned} \quad (22)$$

*If (1) $I(\boldsymbol{Z}_s; \boldsymbol{Z}_c) \leq \epsilon$, and (2) $I(\boldsymbol{Z}_c; Y) \geq \kappa > 0$,*

*then there exist constants $c_1, c_2 > 0$ such that*

$$\Delta_{EO} \leq c_1 \frac{\sqrt{\epsilon}}{\kappa} + c_2 L.$$

**Theorem 3.2.** *(Fair Domain Adaptation Bound)*

*Let $h$ be a classifier on $\boldsymbol{Z}_c$. Then*

$$\begin{aligned} \epsilon_{ta}(h) \leq \epsilon_{so}(h) + C \cdot disc(p_{so}(\boldsymbol{Z}_c), p_{ta}(\boldsymbol{Z}_c)) \\ + \lambda^* + c_1 I(\boldsymbol{Z}_s; \boldsymbol{Z}_c), \end{aligned} \quad (23)$$

*where $disc$ is a discrepancy measure(e.g., $\mathcal{H}\Delta\mathcal{H}$ divergence),$\lambda^*$ is the joint optimal error, and the last term accounts for residual sensitive leakage.*

*Table 1.* Classification and fairness results ($\%\pm\sigma$) on the Bail, German, Pokec, and syn datasets. All values are presented as the mean $\pm$ standard deviation over multiple runs. The symbols ↑ and ↓ indicate that larger and smaller values are preferred, respectively. For clarity, the best-performing result is emphasized using **bold**, whereas the second-highest result is distinguished with an underline.

| Dataset | Metric | GCN | NIFTY | FairVGNN | FairSIN | SFG | SPA | SGDA | FatraGNN | DANCE | CELL |
|---|---|---|---|---|---|---|---|---|---|---|---|
| Bail-t | ACC↑ | 81.37±1.76 | 81.18±0.81 | 82.75±1.54 | 80.1±0.55 | 75.64±13.07 | 84.01±2.71 | 74.40±1.11 | 81.84±2.24 | 84.43±1.85 | **94.21±0.04** |
| | ROC-AUC↑ | 95.39±1.32 | 91.20±0.39 | 91.15±0.87 | 96.28±0.91 | 94.35±0.73 | 89.34±2.94 | 81.41±0.78 | 91.88±0.34 | 95.96±0.49 | **97.96±0.06** |
| | $\Delta_{DP}$ ↓ | 7.37±0.38 | 5.54±0.33 | 11.27±5.55 | 8.07±1.61 | 4.01±1.97 | 5.72±1.45 | 11.50±0.47 | 5.83±0.90 | 3.99±1.05 | **3.88±0.03** |
| | $\Delta_{EO}$ ↓ | 7.73±0.80 | 5.83±0.97 | 10.47±1.77 | **3.75±2.68** | 5.61±2.91 | 4.41±1.92 | 9.68±0.23 | 10.70±0.41 | 4.29±0.69 | 4.98±0.20 |
| | Rank | 5 | 6 | 9 | 3 | 7 | 4 | 8 | 10 | 2 | 1 |
| German-t | ACC↑ | 56.79±3.07 | 56.27±0.79 | 50.51±0.71 | 55.93±3.54 | 54.98±2.11 | 54.03±8.87 | 57.48±2.09 | 56.39±1.83 | **74.29±2.23** | 64.05±0.27 |
| | ROC-AUC↑ | 62.85±1.51 | 65.31±1.03 | 59.36±0.61 | 69.8±1.18 | 63.20±3.61 | 56.04±6.09 | 61.07±1.14 | 65.99±0.25 | 71.63±2.16 | 66.99±0.08 |
| | $\Delta_{DP}$ ↓ | 15.04±13.18 | 9.43±9.01 | 3.65±5.16 | 2.37±2.28 | 7.70±5.35 | 4.33±3.06 | 4.85±1.48 | 18.27±5.45 | 24.76±5.70 | **1.83±1.07** |
| | $\Delta_{EO}$ ↓ | 16.54±13.25 | 9.31±9.63 | 2.90±4.10 | 3.36±1.72 | 7.88±5.41 | 5.35±4.63 | 5.10±3.13 | 17.74±7.44 | 21.89±6.42 | **2.27±1.16** |
| | Rank | 9 | 5 | 4 | 2 | 6 | 7 | 3 | 10 | 8 | 1 |
| Pokec-n | ACC↑ | 68.49±0.33 | 68.08±0.72 | 63.90±2.73 | 63.53±5.84 | 53.61±2.65 | 57.66±1.76 | OOM | 65.22±3.26 | 67.65±0.34 | **68.93±0.20** |
| | ROC-AUC↑ | 76.36±0.24 | 72.68±0.46 | 70.33±0.30 | 70.56±1.03 | 64.02±2.80 | 60.58±2.72 | OOM | 72.66±0.39 | 74.44±0.23 | **76.16±0.11** |
| | $\Delta_{DP}$ ↓ | 2.76±0.42 | 1.79±0.36 | 3.68±1.78 | 3.73±1.74 | 3.31±3.29 | 3.39±1.00 | OOM | 0.93±0.38 | 5.22±0.84 | **0.37±0.30** |
| | $\Delta_{EO}$ ↓ | 2.01±0.44 | 1.98±0.36 | 2.56±1.43 | 3.9±1.8 | 2.47±2.60 | 2.95±1.85 | OOM | 1.46±0.85 | 5.54±0.94 | **0.71±0.33** |
| | Rank | 2 | 3 | 6 | 7 | 9 | 8 | 10 | 4 | 5 | 1 |
| syn-t | ACC↑ | 82.50±0.01 | 82.65±0.13 | 82.55±0.03 | 62.39±9.03 | 84.10±0.83 | 70.58±0.99 | 78.45±0.15 | 81.46±1.59 | **86.87±0.34** | 79.70±0.02 |
| | ROC-AUC↑ | 90.63±0.00 | 90.78±0.03 | 90.58±0.10 | 73.97±7.87 | 90.78±0.40 | 76.34±1.62 | 87.72±0.05 | 90.77±0.01 | **92.36±0.08** | 87.64±0.05 |
| | $\Delta_{DP}$ ↓ | 11.51±0.07 | 12.36±0.27 | 11.01±0.86 | 14.61±10.0 | 26.96±1.44 | 5.14±3.18 | 29.35±0.28 | 13.62±0.05 | 25.34±0.67 | **3.36±0.26** |
| | $\Delta_{EO}$ ↓ | 8.00±0.08 | 9.64±0.52 | 7.44±0.84 | 14.8±10.9 | 30.03±5.88 | 6.72±3.98 | 28.27±1.17 | 3.17±0.03 | 23.13±1.00 | **0.42±0.27** |
| | Rank | 3 | 4 | 2 | 9 | 8 | 6 | 10 | 7 | 5 | 1 |

**Lemma 3.3.** *(Bias Control with Class-wise Thresholds).*

*Let per-class adaptive thresholds $\tau_k = M_k\tau$ with $M_k = max\{m_v^{ta} : arg\,max\,\psi(\boldsymbol{z}_{v,c}) = k\}$. Define confident set $C = \{v : m_v^{ta} > \tau_k\}$. Then selection bias satisfies*

$$Bias_{sel} = \sum_k |Pr(v \in C|Y = k) - \rho| \qquad (24)$$
$$\leq \sum_k |Pr(m_v^{ta} > \tau_k|Y = k) - \rho|,$$

*for target coverage $\rho$. Adaptive $\tau_k$ balances coverage across classes, reducing bias while training on confident samples.*

## 4. Experiments

**Datasets & Baselines.** We evaluate CELL on three enhanced real-world graphs and one synthetic benchmark from Qian et al. (Qian et al., 2024): 1) *Credit-Cs* is built upon the Credit dataset (Yeh & Lien, 2009). 2) *Pokecs* comes from a Slovak social network (Dai & Wang, 2022). 3) *Bail-Bs* is derived from the Bail dataset (Jordan & Freiburger, 2015), where nodes are defendants released on bail. 4) *Synthetic* test fair GNNs when edges carry signal and topology can amplify bias (Qian et al., 2024). Our CELL is compared with four types of baselines: (A) Traditional learning approaches, including GCN (Kipf & Welling, 2017). (B) Fairness-Aware GNNs under independent and identically distributed (IID) settings, including NIFTY (Agarwal et al., 2021), FairVGNN (Wang et al., 2022), FairSIN (Yang et al., 2024), and SFG (Chen et al.). (C) General Domain Adaptation Methods, including SGDA (Qiao et al., 2023) and SPA (Xiao et al., 2023). (D) Fairness-Aware GNNs under Out-of-Distribution (OOD) Settings, including FatraGNN (Li et al.,

2024) and DANCE (Wang et al., 2025). Additional details on the experimental settings can be found in Appendix B.

**Performance Evaluation.** We measure node classification performance with two commonly used indicators, namely accuracy (ACC) and ROC_AUC. Model fairness is quantified by $\Delta_{DP}$ and $\Delta_{EO}$, as introduced in Section 2, where smaller values of these two metrics correspond to fairer predictions. To jointly account for predictive performance and fairness, we further define an aggregate criterion as $c = \text{ACC} + \text{ROC\_AUC} - \Delta_{DP} - \Delta_{EO}$, where larger values denote a better balance between predictive utility and fairness. For each method, its final result is computed by accumulating the scores obtained on the target domains, and the methods are ranked based on these overall scores.

### 4.1. Performance Analysis

Table 1 summarizes the best average results achieved by all compared methods on the four datasets. From these results, we can make several observations. First, compared with conventional learning approaches, fairness-aware GNN models generally achieve better fairness metrics, but this improvement is often accompanied by a decline in classification accuracy. Moreover, because Fair GNNs are primarily tailored to IID settings, their performance tends to degrade in domain adaptation tasks, where they can even achieve a less favorable accuracy-fairness trade-off than vanilla GCNs. (2) When comparing Fair GNNs with fairness-aware GNNs designed for OOD settings, we observe that OOD methods achieve a better balance between classification and fairness under domain adaptation, owing to their ability to learn fair representations across varying distributions. This highlights

*Table 2.* Ablation Studies on CELL Variants.

| Variant | Pokec-n | | | | syn-t | | | |
|---|---|---|---|---|---|---|---|---|
| Metric | ACC↑ | ROC-AUC↑ | $\Delta_{DP}\downarrow$ | $\Delta_{EO}\downarrow$ | ACC↑ | ROC-AUC↑ | $\Delta_{DP}\downarrow$ | $\Delta_{EO}\downarrow$ |
| Var1 | **69.79±0.18** | **76.97±0.21** | 0.73±0.26 | **0.26±0.08** | 77.83±4.51 | 85.46±4.91 | 9.01±4.85 | 7.65±6.41 |
| Var2 | 69.14±0.15 | 75.85±0.15 | 0.79±0.38 | 0.93±0.50 | 79.47±0.15 | 87.47±0.05 | **2.60±0.17** | 1.47±0.33 |
| Var3 | 69.07±0.11 | 75.78±0.17 | 0.95±0.08 | 1.17±0.17 | 80.43±0.04 | 88.46±0.04 | 5.85±0.14 | 2.22±0.22 |
| Var4 | 68.66±0.13 | 75.86±0.04 | 1.22±0.04 | 1.01±0.06 | **81.56±0.05** | **89.81±0.02** | 10.30±0.43 | 7.25±0.36 |
| CELL | 68.93±0.20 | 76.16±0.11 | **0.37±0.30** | 0.71±0.33 | 79.70±0.02 | 87.64±0.05 | 3.36±0.26 | **0.42±0.27** |

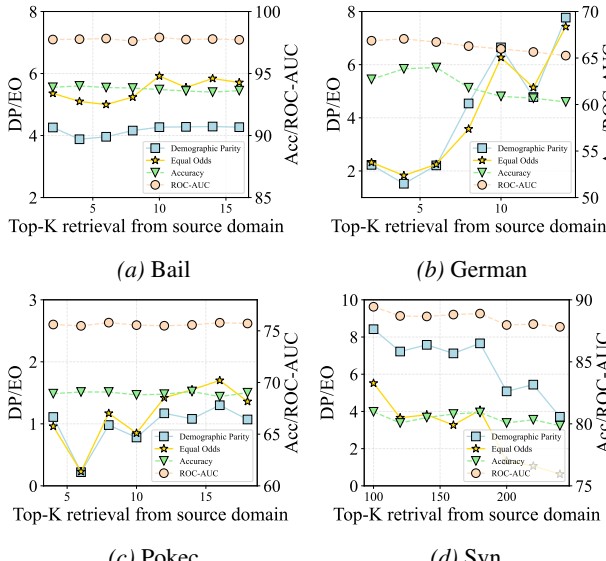

*(a)* Bail      *(b)* German

*(c)* Pokec      *(d)* Syn

*Figure 2.* Utility and fairness comparison w.r.t. different top-$K$ for retrieval nodes from source domain.

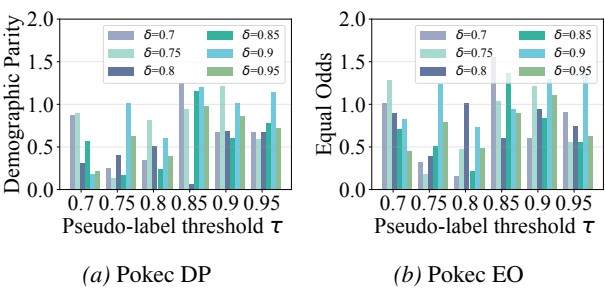

*(a)* Pokec DP      *(b)* Pokec EO

*Figure 3.* Utility and fairness comparison w.r.t. different values for threshold parameter $\tau$ and $\delta$.

the necessity of incorporating domain adaptation modules in fairness-aware GNNs. (3) When comparing fairness-aware GNNs under OOD settings with general domain adaptation methods, we find that the latter often achieve either high performance or strong fairness, but struggle to balance both. This underscores the scarcity of fairness-aware GNN approaches specifically designed for domain adaptation scenarios. (4) CELL outperforms all other baselines in most cases. Compared to fairness-aware GNNs under IID and OOD settings, as well as general domain adaptation methods, CELL strikes a better balance between accuracy and fairness in domain adaptation, demonstrating the effective-

ness of its domain adaptation module design.

## 4.2. Ablation Study

To quantify the contribution of each component in CELL, we evaluate four ablated variants: (1) Variant 1: Replace the MI-based GNN encoder with a vanilla GCN. (2) Variant 2: Remove the bipartite-aware domain alignment module. (3) Variant 3: Remove the sensitive consistency learning module. (4) Variant 4: Remove pseudo-label supervision and rely only on source ground-truth labels. Table 2 summarizes the results, from which we draw four key observations. (1) Compared to Variant 1, substituting the MI-based GNN with a vanilla GCN substantially degrades both utility and fairness. The MI objective helps learn fairness-aware embeddings that support reliable cross-domain top-$k$ retrieval. Without MI-based GNN, sensitive information leaks into representations, retrieval becomes biased, and downstream alignment and consistency training are less effective. (2) Compared to Variant 2, incorporating bipartite-aware alignment by explicitly linking same-label source–target nodes and regularizing $PP^T$ shrinks the cross-domain discrepancy. This improves pseudo-label stability, limits biased propagation through cross-domain edges, and enhances both accuracy and fairness. (3) Compared to Variant 3, removing the sensitive consistency loss leads to worse fairness and a slight drop in utility. Enforcing sensitive inter-group invariance and appropriate separation in the target embedding space mitigates sensitive-attribute leakage and maintains better cross-domain representation consistency. (4) Compared to Variant 4, removing pseudo-labels harms both utility and fairness. High-confidence pseudo-labels provide essential target-side supervision to adapt the decision boundary, define label-consistent alignment pairs and cleaner cross-domain top-$k$ retrieval, and activate the sensitive consistency objective on target nodes. Without them, optimization is driven solely by source supervision, cross-domain pairing becomes noisier, and representation drift increases bias.

## 4.3. Parameter Analysis

Here, we study the influence of three hyperparameter groups in CELL: (1) the number of top-$K$ source nodes retrieved by the MI-based GNN, (2) the pseudo-label confidence

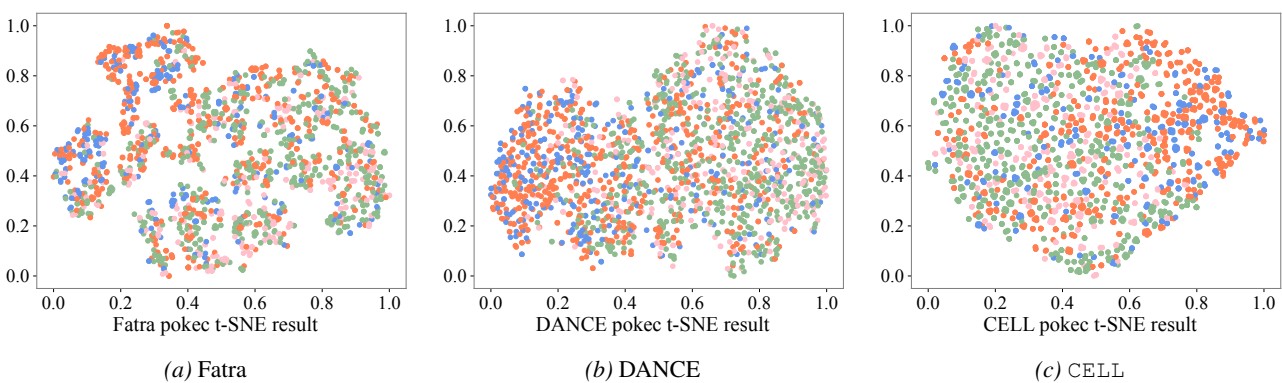

*(a)* Fatra           *(b)* DANCE           *(c)* CELL

*Figure 4.* Domain adaptation t-SNE: Source-Learned Representations on Pokec-n (target domain)

thresholds $\tau$ and $\delta$, and (3) the loss weights $\beta$ and $\gamma$ in the objective function Eq. 21. Our key findings are as follows: (1) As shown in Figures 2a and 2b, too small top-$K$ underutilizes informative cross-domain connections, while an overly large $K$ introduces noisy and potentially demographically imbalanced neighbors. Both effects typically degrade utility and fairness on most datasets. The optimal $K$ is clearly dataset-dependent, varying substantially across Bail, German, Pokec, and syn. (2) As shown in Figure 3a and 3b, moderate–high thresholds improve both utility and fairness by filtering unreliable pseudo-labels and sharpening the effective decision boundary. In practice, effective regions are often observed at $\delta, \tau \geq 0.75$, although the optimal values differ by dataset. Low thresholds admit excessive noise and tend to harm fairness, whereas overly strict thresholds reduce the amount of usable supervision and may lower utility. (3) As shown in Figure 5, the optimal $(\gamma, \beta)$ configurations vary widely across Bail, German, Pokec, and syn, and no single setting consistently dominates in terms of fairness. Some datasets benefit from stronger alignment (larger $\gamma$) with weaker fairness regularization (smaller $\beta$), while others exhibit the opposite pattern. This variability suggests that the correlations among labels, sensitive attributes, and graph structure differ across datasets, making dataset-specific weighting necessary to balance predictive performance and group fairness.

### 4.4. Visualization

We visualize source-trained embeddings on the Pokec-n target graph using t-SNE (Van der Maaten & Hinton, 2008), with points colored by the $(Y, S)$ groups (task and sensitive labels). As shown in Fig. 4, Fatra exhibits multiple well-separated islands, indicating that the learned representation is strongly aligned with group identity. DANCE promotes greater inter-group mixing, but clear cluster boundaries remain. In contrast, CELL yields the most group-intermixed manifold with fewer isolated clusters, indicating reduced sensitive-attribute leakage and improved cross-domain alignment. Although t-SNE provides only qualitative evidence,

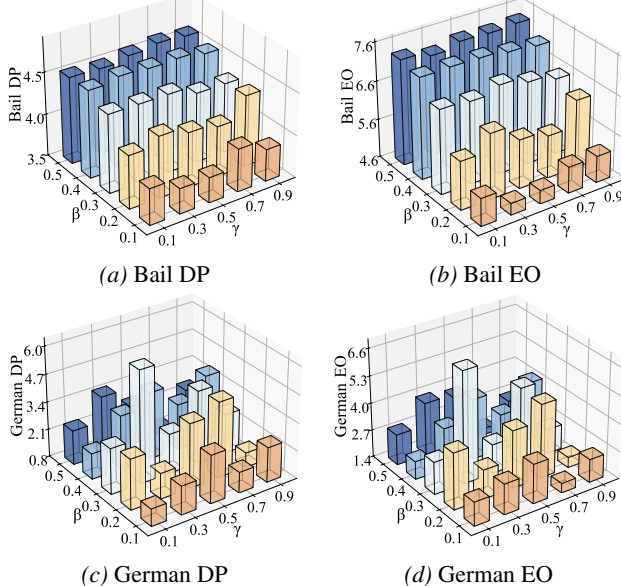

*(a)* Bail DP           *(b)* Bail EO

*(c)* German DP           *(d)* German EO

*Figure 5.* Fairness comparison w.r.t. different values for loss weights $\gamma$ and $\beta$

these trends are consistent with the quantitative results (lower DP and EO) while maintaining competitive utility.

## 5. Conclusion

In this paper, we propose CELL, a causality-attended representation disentanglement and structural alignment framework for fairness-aware graph domain adaptation. Beyond i.i.d. assumptions, CELL separates task and sensitive factors using dual encoders with mutual-information–based disentanglement, enhances target-domain supervision via group-aware pseudo-labeling, and mitigates cross-domain spurious correlations through fairness-aware bipartite alignment. Experiments on three real-world and one synthetic benchmark show improvements in both predictive utility and group fairness under distribution shifts, positioning CELL as a practical and reliable solution for fair graph learning in real-world cross-domain settings.

## Impact Statement

Our CELL framework improves fairness-aware graph adaptation under domain shifts by transferring predictive and fairness knowledge from a labeled source graph to a fully unlabeled target graph, without requiring target sensitive labels. CELL adopts a causal view of graph generation and disentangles task-relevant factors from sensitive-related factors via a dual-encoder with mutual-information constraints, preserving predictive signals while reducing sensitive leakage. CELL then exploits unlabeled target data through pseudo-labeling and a group-attended unbiased learning strategy that refines intra-/inter-group similarities to encourage fair representations. Finally, a fairness-aware bipartite alignment links cross-domain nodes with the same task labels but different sensitive attributes, narrowing domain gaps and mitigating spurious correlations. Together, these components yield stronger out-of-distribution generalization and improved group fairness, with practical relevance to settings like credit risk prediction and criminal justice decision support where shifts and missing demographic labels are common.

## Acknowledgement

Ming Zhang is supported by the National Key Research and Development Program of China under Grant No. 2023YFC3341203 and the National Natural Science Foundation of China under Grant No. 62276002. Wei Ju is supported by the National Natural Science Foundation of China under Grant No. 62306014. Yifan Wang is supported by the Fundamental Research Funds for the Central Universities in UIBE under Grant No. 23QN02 and the Humanities and Social Sciences Research Fund of the Ministry of Education of China under Grant No. 25YJCZH275.

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

# A. Related Work

**Group Fairness in GNNs.** Graph neural networks (GNNs) (Wang et al., 2024) can inadvertently propagate bias because sensitive attributes are often entangled with graph topology (Chen et al., 2024). Fairness in GNNs is typically studied along two axes, namely group fairness and individual fairness, with most work targeting the former by equalizing outcomes across demographic subpopulations. A prominent line for group fairness removes sensitive information from representations via adversarial learning: FairGNN (Dai & Wang, 2021) trains an auxiliary discriminator and adds a covariance penalty to encourage statistical parity, while FairVGNN (Wang et al., 2022) employs a discriminator to hide protected attributes in learned representations. Beyond adversarial debiasing, explicit group-fairness objectives are optimized by jointly minimizing accuracy loss with disparity-based penalties using Lagrangian or multi-objective formulations (Chen et al., 2024). For example, NIFTY constructs counterfactual feature views by flipping the sensitive attribute and applies a contrastive loss that pulls original–counterfactual pairs together while pushing apart label-inconsistent pairs (Agarwal et al., 2021). Another complementary line de-biases input structure: For example, FairWalk (Rahman et al., 2019) reweights random walks to yield demographically balanced contexts EDITS (Dong et al., 2022) removes sensitive cues from node features while preserving utility. In summary, these research avenues outline mainstream paradigms—adversarial, constraint-based, contrastive, and structural—for mitigating bias in GNN-based learning.

**Fairness under Distribution Shift.** Distribution shift can significantly degrade fairness when test and training distributions diverge (Liu et al., 2021; An et al., 2022). In response, a growing literature seeks to maintain fairness amidst such shifts. (Rezaei et al., 2021; Giguere et al., 2022; Lin et al., 2024). (Rezaei et al., 2021) propose a robust-fairness method for covariate shift that adapts models using unlabeled target-domain data. (Giguere et al., 2022) focus on demographic shift—changes in subgroup prevalences—and provide high-confidence fairness guarantees when test-time group proportions differ from training. (Mandal et al., 2020) adopt a worst-case, distributionally robust approach, modeling the test distribution as a weighted combination of training samples and optimizing fairness under this adversarial shift. For a more full review for fairness under distribution shifts, please refer to a recent survey (Lin et al., 2024). The majority of these approaches ignore relational structure and assume Euclidean data. In order to handle distribution shifts, FatraGNN (Li et al., 2024) explicitly addresses graphs by producing additional biased training graphs and minimizing group-wise representation distances between the created and original graphs. In addition, DANCE (Wang et al., 2025) addresses group imbalance and emphasizes fairness under graph shifts, enhancing fairness under shifting distributions without compromising task performance.

# B. Detailed Experimental Settings

**Datasets.** We conduct experiments on three augmented real-world graph datasets and one synthetic benchmark adopted from Qian et al. (Qian et al., 2024):

1) *Credit-Cs* is generated based on the Credit dataset (Yeh & Lien, 2009), where each node corresponds to a credit card holder. The objective is to perform binary credit-risk classification, and *age* is treated as the sensitive attribute. Following a modularity-based community detection strategy (Newman, 2006), the graph is partitioned into *credit-s* and *credit-t*, which serve as the source and target domains, respectively, with different data distributions. 2) *Pokecs* originates from a Slovak social network dataset (Dai & Wang, 2022), in which users are organized according to their provinces. The prediction task is to infer users' occupations or working fields, while *region* is used as the sensitive attribute. This dataset contains two graph domains, namely *Pokec-z* and *Pokec-n*. The former is adopted as the source domain and the latter as the target domain. 3) *Bail-Bs* is constructed from the Bail dataset (Jordan & Freiburger, 2015), where nodes denote defendants released on bail. The learning task is to predict whether bail should be granted, with *race* regarded as the sensitive attribute. Similar to the construction of Credit-Cs, modularity-based community detection is applied to divide the graph into *Bail-s* and *Bail-t*, which are used as the source and target domains, respectively. 4) *Synthetic* test fair GNNs when edges carry signal and topology can amplify bias (Qian et al., 2024). For each node, $(S, Y)$ are jointly sampled from a categorical distribution with user-specified group proportions. Features concatenate two multivariate Gaussians conditioned on $S$ and $Y$ with tunable means/variances. Each edge type is generated independently via its own Bernoulli probability. We use *Syn-2* as source domain and *Syn-1* as target domain.

**Baselines.** Our `CELL` is compared with four types of baselines: (A) Traditional learning approaches: Fundamental graph representation learning approaches, including GCN (Kipf & Welling, 2017). (B) Fairness-Aware GNNs under independent and identically distributed (IID) settings: GNNs specifically designed to enhance fairness in IID scenarios, including NIFTY (Agarwal et al., 2021), FairVGNN (Wang et al., 2022), FairSIN (Yang et al., 2024), and SFG (Chen et al.). (C) General

*Table 3.* Ablation studies on the variants of CELL.

| Variant | Bail-t | | | | German-t | | | |
|---|---|---|---|---|---|---|---|---|
| Metric | ACC↑ | ROC-AUC↑ | $\Delta_{DP} \downarrow$ | $\Delta_{EO} \downarrow$ | ACC↑ | ROC-AUC↑ | $\Delta_{DP} \downarrow$ | $\Delta_{EO} \downarrow$ |
| Var1 | 92.58±0.95 | 97.06±0.26 | **3.42±0.75** | 5.45±0.40 | 63.26±0.08 | **67.34±0.12** | 5.14±0.80 | 9.75±0.26 |
| Var2 | 93.72±0.18 | 97.81±0.10 | 4.16±0.08 | 5.59±0.14 | **64.11±0.41** | 67.04±0.13 | 2.77±1.58 | 3.10±1.65 |
| Var3 | 93.81±0.03 | 97.66±0.08 | 4.09±0.04 | 5.06±0.25 | 63.63±0.23 | 66.92±0.15 | 2.65±1.60 | 3.38±1.71 |
| Var4 | 93.78±0.13 | 97.83±0.08 | 4.12±0.03 | 5.37±0.18 | 63.75±0.62 | 66.99±0.09 | 2.22±1.46 | 2.30±1.16 |
| CELL | **94.21±0.04** | **97.96±0.06** | 3.88±0.03 | **4.98±0.20** | 64.05±0.27 | 66.99±0.08 | **1.83±1.07** | **2.27±1.16** |

*Table 4.* Ablation studies on the variants of CELL.

| Variant | Pokec-n | | | | syn-t | | | |
|---|---|---|---|---|---|---|---|---|
| Metric | ACC↑ | ROC-AUC↑ | $\Delta_{DP} \downarrow$ | $\Delta_{EO} \downarrow$ | ACC↑ | ROC-AUC↑ | $\Delta_{DP} \downarrow$ | $\Delta_{EO} \downarrow$ |
| Var1 | **69.79±0.18** | **76.97±0.21** | 0.73±0.26 | **0.26±0.08** | 77.83±4.51 | 85.46±4.91 | 9.01±4.85 | 7.65±6.41 |
| Var2 | 69.14±0.15 | 75.85±0.15 | 0.79±0.38 | 0.93±0.50 | 79.47±0.15 | 87.47±0.05 | **2.60±0.17** | 1.47±0.33 |
| Var3 | 69.07±0.11 | 75.78±0.17 | 0.95±0.08 | 1.17±0.17 | 80.43±0.04 | 88.46±0.04 | 5.85±0.14 | 2.22±0.22 |
| Var4 | 68.66±0.13 | 75.86±0.04 | 1.22±0.04 | 1.01±0.06 | **81.56±0.05** | **89.81±0.02** | 10.30±0.43 | 7.25±0.36 |
| CELL | 68.93±0.20 | 76.16±0.11 | **0.37±0.30** | 0.71±0.33 | 79.70±0.02 | 87.64±0.05 | 3.36±0.26 | **0.42±0.27** |

Domain Adaptation Methods: General approaches aimed at learning robust representations for domain adaptation, including SGDA (Qiao et al., 2023) and SPA (Xiao et al., 2023). (D) Fairness-Aware GNNs under Out-of-Distribution (OOD) Settings: Graph neural network methods specifically designed to address distribution shifts while ensuring fairness between training and test distributions, such as FatraGNN (Li et al., 2024) and DANCE (Wang et al., 2025).

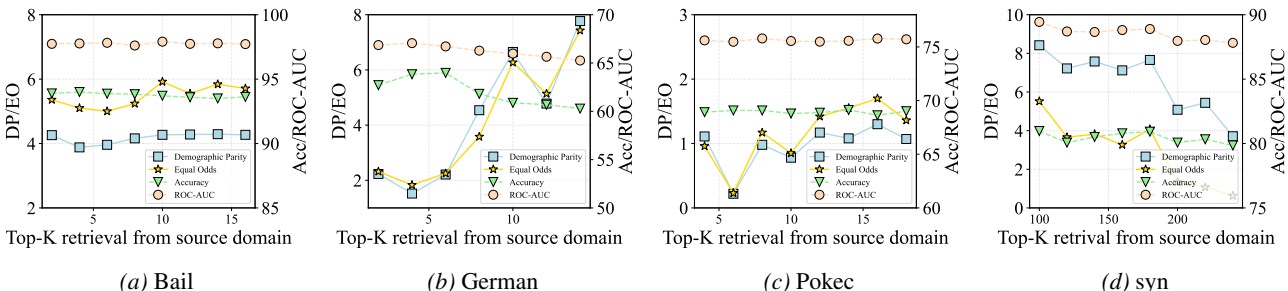

*Figure 6.* Utility and fairness comparison w.r.t. different top-$K$ for retrieval node from source domain.

## C. Additional Results

### C.1. Experimental Setting.

In our experiments, hyperparameters are selected through grid search over all dataset groups, aiming to provide a fair and thorough comparison. For CELL, we fix the embedding size to 64. The number of graph encoder layers is searched within $[2, 4]$, the dropout ratio is varied from 0 to 0.5, and the learning rate is chosen from $[0.002, 0.006]$. To reduce the influence of randomness, we repeat each method five times using different random seeds and report the mean value together with the variance for each evaluation metric.

### C.2. More ablation study results

The results of the complete ablation studies are presented in Tables 3 and 4.

### C.3. More parameter analysis results

The results of the complete parameter analysis are presented in Figures 6, 7 and 8.

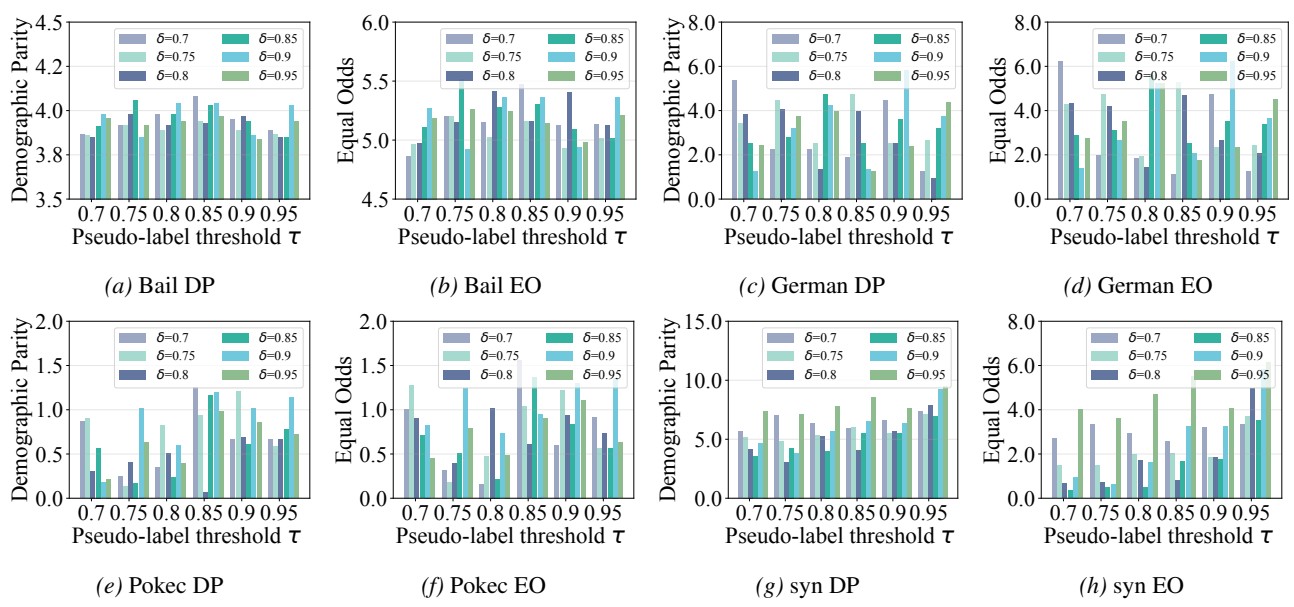

*Figure 7.* Fairness comparison w.r.t. different values for threshold parameter $\tau$ and $\delta$

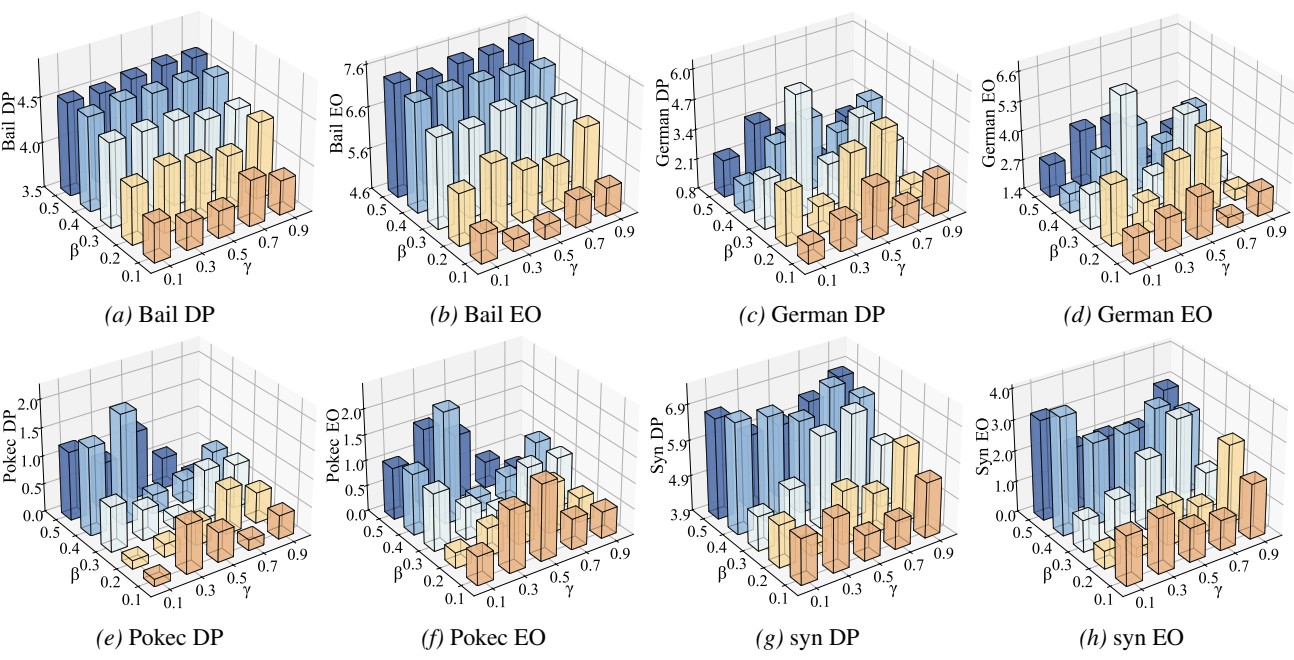

*Figure 8.* Fairness comparison w.r.t. different values for loss weights $\gamma$ and $\beta$

## D. Proof of Theoretical Results

**Theorem D.1.** *(CLUB Upper Bound for $I(\mathbf{Z}_s; \mathbf{Z}_c)$).*

*For any conditional density estimator $q_\theta(\boldsymbol{z}_s|\boldsymbol{z}_c)$, define*

$$CLUB_\theta := \mathbb{E}_{p(\boldsymbol{z}_s|\boldsymbol{z}_c)}[logq_\theta(\boldsymbol{z}_s|\boldsymbol{z}_c)] - \mathbb{E}_{p(\boldsymbol{z}_c)p(\boldsymbol{z}_s)}[logq_\theta(\boldsymbol{z}_s|\boldsymbol{z}_c)]$$

*Then,*

$$I(\boldsymbol{Z}_s; \boldsymbol{Z}_c) \leq CLUB_\theta + \mathbb{E}_{p(\boldsymbol{z}_s, \boldsymbol{z}_c)} log \frac{p(\boldsymbol{z}_s|\boldsymbol{z}_c)}{q_\theta(\boldsymbol{z}_s|\boldsymbol{z}_c)}.$$

*Hence, $CLUB_\theta$ is an upper bound on $I(\boldsymbol{Z}_s; \boldsymbol{Z}_c)$; the bound tightens as $q_\theta \to p(\cdot|\cdot)$.*

*Proof.* By definition, $I(\boldsymbol{Z}_s; \boldsymbol{Z}_c) = \mathbb{E}_{p(\boldsymbol{z}_s, \boldsymbol{z}_c)}[\log \frac{p(\boldsymbol{z}_s|\boldsymbol{z}_c)}{p(\boldsymbol{z}_s)}]$. For any variational distribution $q_\theta$, we add and subtract $\log q_\theta(\boldsymbol{z}_s, \boldsymbol{z}_c)$:

$$I(\mathbf{Z}_s; \mathbf{Z}_c) = \Big(\mathbb{E}_{p(\mathbf{z}_s, \mathbf{z}_c)} \log q_\theta(\mathbf{z}_s \mid \mathbf{z}_c) - \mathbb{E}_{p(\mathbf{z}_c)p(\mathbf{z}_s)} \log q_\theta(\mathbf{z}_s \mid \mathbf{z}_c)\Big) + \mathbb{E}_{p(\mathbf{z}_s, \mathbf{z}_c)} \left[\log \frac{q_\theta(\mathbf{z}_s \mid \mathbf{z}_c)}{p(\mathbf{z}_s \mid \mathbf{z}_c)}\right]$$

$$= \text{CLUB}_\theta + \mathbb{E}_{p(\mathbf{z}_c)} \text{KL}\big(p(\cdot \mid \mathbf{z}_c) \,\|\, q_\theta(\cdot \mid \mathbf{z}_c)\big).$$

Since the KL term is always non-negative, the inequality follows. Equality holds when $q_\theta = p$. $\square$

**Theorem D.2.** *(Conditional InfoNCE as a Lower Bound on $I(\mathbf{Z}_C; D|\mathbf{Z}_s)$)*

*Let $f(\boldsymbol{\mu}, \boldsymbol{z}_c, \boldsymbol{z}_s)$ be a bounded scoring function. For batch size $M$, define*

$$\mathcal{L}_{cNCE} := -\mathbb{E}[log \frac{exp f(\boldsymbol{\mu}_v, \boldsymbol{z}_{v,c}, \boldsymbol{z}_{v,s})}{\frac{1}{M} \sum_{u=1}^{M} exp f(\boldsymbol{\mu}_u, \boldsymbol{z}_{v,c}, \boldsymbol{z}_{v,s})}].$$

*Then $I(\mu; \boldsymbol{Z}_c|\boldsymbol{Z}_s) \geq -\mathcal{L}_{cNCE}$. If the anchor $\boldsymbol{\mu}$ encodes domain-related information, there exists $\kappa > 0$ such that $I(\boldsymbol{Z}_c; D|\boldsymbol{Z}_s) \geq -\kappa\mathcal{L}_{cNCE}$.*

*Proof.* Given $C = c$, build a contrastive set $\mathcal{S} = \{(\boldsymbol{\mu}_u, \mathbf{z}_{u,c})\}_{u=1}^{M}$ containing one positive from $p_+(\cdot \mid c)$ and $M-1$ negatives from $p_-(\cdot \mid c)$. Let $K \in \{1, \ldots, M\}$ be the (uniform) index of the positive pair. Define the softmax classifier

$$q_f(k \mid \mathcal{S}, c) = \frac{\exp f(\boldsymbol{\mu}_k, \mathbf{z}_{k,c}, c)}{\sum_{i=1}^{M} \exp f(\boldsymbol{\mu}_i, \mathbf{z}_{i,c}, c)}. \tag{25}$$

The usual sum-denominator InfoNCE cross-entropy(van den Oord et al., 2018) is $\mathcal{L}_{\text{sum}} = \mathbb{E}[-\log q_f(K \mid \mathcal{S}, C)]$, and our avg loss satisfies the identity $\mathcal{L}_{\text{cNCE}} = \mathcal{L}_{\text{sum}} - \log M$. At the Bayes score $f^\star = \log \frac{p_+}{p_-}$, $q_{f^\star} = p(K \mid \mathcal{S}, C)$, hence

$$\min_f \mathcal{L}_{\text{sum}} = \mathbb{E}[-\log p(K \mid \mathcal{S}, C)] = \log M - I(K; \mathcal{S} \mid C), \tag{26}$$

since $H(K \mid C) = \log M$ and $I(K; \mathcal{S} \mid C) = H(K \mid C) - H(K \mid \mathcal{S}, C)$. For any $f$ (by Gibbs' inequality), $\mathcal{L}_{\text{sum}} \geq \log M - I(K; \mathcal{S} \mid C)$. By data processing in the sampling scheme (the only dependence of $K$ on $\mathcal{S}$ flows through the joint information in $(\boldsymbol{\mu}, \mathbf{Z}_c)$), $I(K; \mathcal{S} \mid C) \leq I(\boldsymbol{\mu}; \mathbf{Z}_c \mid C)$. Thus

$$\mathcal{L}_{\text{sum}} \geq \log M - I(\boldsymbol{\mu}; \mathbf{Z}_c \mid C) \Longleftrightarrow I(\boldsymbol{\mu}; \mathbf{Z}_c \mid C) \geq \log M - \mathcal{L}_{\text{sum}}. \tag{27}$$

Using $\mathcal{L}_{\text{cNCE}} = \mathcal{L}_{\text{sum}} - \log M$ gives the stated avg-form bound: $I(\boldsymbol{\mu}; \mathbf{Z}_c \mid C) \geq -\mathcal{L}_{\text{cNCE}}$. $\square$

**Theorem D.3.** *(Fairness Upper Bound)*

*Let classifier $h$ depend only on $\boldsymbol{Z}_c$. Assume $h$ is $L$-lipschitz in distribution shift. Define*

$$\Delta_{EO} = \sum_{y \in \{0,1\}} |P(h = 1|S = 0, Y = y) - P(h = 1|S = 1, Y = y)|$$

*If (1) $I(\boldsymbol{Z}_s; \boldsymbol{Z}_c) \leq \epsilon$, and (2) $I(\boldsymbol{Z}_c; Y) \geq \kappa > 0$,*

*then there exist constants $c_1, c_2 > 0$ such that*

$$\Delta_{EO} \leq c_1 \frac{\sqrt{\epsilon}}{\kappa} + c_2 L.$$

*Proof.* Fix $y \in \{0,1\}$ and let $P_y := \mathcal{L}(\boldsymbol{Z}_c|S=0, Y=y)$, $Q_y = \mathcal{L}(\boldsymbol{Z}_c|S=1, Y=y)$. Let $A_h := \{\boldsymbol{z}_c : h(\boldsymbol{z}_c) = 1\}$. Then $|\Pr(h=1|S=0, Y=y) - \Pr(h=1|S=1, Y=y)| = |P_y(A_h) - Q_y(A_h)| \leq \mathrm{TV}(P_y, Q_y)$: hence

$$\Delta_{\mathrm{EO}} \leq \sum_{y \in \{0,1\}} \mathrm{TV}(P_y, Q_y). \tag{28}$$

For binary $S$, standard inequalities(Pinsker's inequality(Pinsker, 1964)) relating TV and KL plus the identity $I(\boldsymbol{Z}_s; S|Y = y) = \sum_{s \in \{0,1\}} \Pr(s|y) \mathrm{KL}(\mathcal{L}(\boldsymbol{Z}_c|s,y)||\mathcal{L}(\boldsymbol{Z}_c|y))$, give(absorbing the class-imbalance factor into a constant $C_y$)

$$\mathrm{TV}(P_y, Q_y) \leq C_y \sqrt{I(\boldsymbol{Z}_c; S|Y = y)}. \tag{29}$$

Summing over $y$ and using $\sqrt{a} + \sqrt{b} \leq \sqrt{2(a+b)}$ yields

$$\sum_y \mathrm{TV}(P_y, Q_y) \leq C \sqrt{I(\boldsymbol{Z}_c; S|Y)} \tag{30}$$

for a constant $C > 0$ that depends only on label/group proportions.

Since $\boldsymbol{Z}_s$ is a proxy for $S$, by data processing and a near-sufficiency argument there exists a small $\delta \geq 0$ such that

$$I(\boldsymbol{Z}_c; S|Y) \leq I(\boldsymbol{Z}_c; \boldsymbol{Z}_s|Y) + \delta \leq I(\boldsymbol{Z}_c; \boldsymbol{Z}_s) + \delta \leq \epsilon + \delta. \tag{31}$$

So $\Delta_{\mathrm{EO}} \leq C\sqrt{\epsilon + \delta}$.

Assumption $I(\boldsymbol{Z}_c; Y) \geq \kappa$ says $\boldsymbol{Z}_c$ carries at least $\kappa$ nats of task signal, Standard stability/margin arguments (Bartlett et al., 2006; Shalev-Shwartz et al., 2010; Bousquet & Elisseeff, 2002)imply that the contribution of spurious variation in $\boldsymbol{Z}_c$ to the decision probability is attenuated by a factor proportional to $\frac{1}{\kappa}$.Thus there is a constant $c_1 > 0$, such that

$$\Delta_{\mathrm{EO}} \leq c_1 \frac{\sqrt{\epsilon + \delta}}{\kappa} + c_2 L, \tag{32}$$

where the additive $c_2 L$ term accounts for the $L$-Lipschitz sensitivity of $h$ under residual distributional shift not captured by the MI control(a standard device to prevent amplification when mapping input distributions to predictions).

Finally, absorb $\delta$ into $\epsilon$ and rename constants to get

$$\Delta_{\mathrm{EO}} \leq c_1 \frac{\sqrt{\epsilon}}{\kappa} + c_2 L \tag{33}$$

. as claimed. $\square$

**Lemma D.4.** *(Spectral Consistency)*

*Let bipartite adjacency $B$ connect source-target pairs with identical task labels but different sensitive labels. Define normalized Laplacian $L$ and prediction matrix $P$.Then*

$$L_{ba} = ||(I - L) - PP^T||_F^2 = -2 \sum_{u,v} \frac{B_{uv}}{\sqrt{d_u d_v}} \boldsymbol{p}_u^T \boldsymbol{p}_v + \sum_{u,v} B_{uv}(\boldsymbol{p}_u^T \boldsymbol{p}_v)^2 + \text{const.}$$

*Proof.* Expand the Frobenius norm:

$$\begin{aligned} L_{\mathrm{ba}} &= \|\mathbf{A}\|_F^2 + \|\mathbf{PP}^\top\|_F^2 - 2\,\mathrm{tr}(\mathbf{A}^\top \mathbf{PP}^\top) \\ &= \|\mathbf{A}\|_F^2 + \|\mathbf{PP}^\top\|_F^2 - 2\,\mathrm{tr}(\mathbf{P}^\top \mathbf{AP}). \end{aligned} \tag{34}$$

(i) For the trace term:

$$\mathrm{tr}(\mathbf{P}^\top \mathbf{AP}) = \sum_{u,v} A_{uv} \mathbf{p}_u^\top \mathbf{p}_v = \sum_{u,v} \frac{B_{uv}}{\sqrt{d_u d_v}} \mathbf{p}_u^\top \mathbf{p}_v. \tag{35}$$

(ii) For the squared term:

$$\|\mathbf{P}\mathbf{P}^\top\|_F^2 = \sum_{u,v} \left((\mathbf{P}\mathbf{P}^\top)_{uv}\right)^2 = \sum_{u,v} (\mathbf{p}_u^\top \mathbf{p}_v)^2. \tag{36}$$

If we only accumulate over edges $(u, v)$ with $B_{uv} = 1$ (as in Eq. (16) of the main text).

(iii) Under the constraint $\mathbf{P}^\top \mathbf{P} = \mathbf{I}_K$, we have $\|\mathbf{P}\mathbf{P}^\top\|_F^2 = K$ is constant. Therefore minimizing $L_{ba}$ is equivalent to maximizing $\mathrm{tr}(\mathbf{P}^\top \mathbf{A}\mathbf{P})$, which by the Rayleigh–Ritz theorem (Horn & Johnson, 2012)is achieved by choosing $\mathbf{P}$'s columns as the top $K$ eigenvectors of $\mathbf{A}$. Equivalently, these correspond to the bottom $K$ eigenvectors of $\mathbf{L}$, consistent with standard spectral clustering. □

**Lemma D.5.** *(Effect of Group-wise Contrastive Losses).*

*Partition neighbors into IG/SG/TG/TSG sets. Define losses $L_{scl}$ and $L_{dis}$ as in the main text. Then:*

*(1)Minimizing $L_{scl}$ increases a lower bound of $I(\mathbf{Z}_c; Y)$ by treating IG and SG as positives, TG as negatives.*

*(2)Minimizing $L_{dis}$ reduces effective $I(\mathbf{Z}_c; \mathbf{Z}_s)$ by penalizing alignment between task and sensitive embeddings.*

*Proof.* Let $Z_c$ denote task representations and $Z_s$ denote sensitive representations. For each anchor $i$ with label $Y_i$, partition neighbors into IG/SG/TG/TSG; in the contrastive loss $L_{scl}$ we treat IG∪SG as positives and TG as negatives (TSG may be folded into TG as hard negatives). With a similarity score $\mathrm{sim}(\cdot, \cdot)$, $L_{scl}$ follows the InfoNCE form where one positive $j^+$ is sampled from an approximation of the class-conditional $p(z^c \,|\, Y_i)$ and $N-1$ negatives $\{j^-\}$ are sampled from the marginal $p(z^c)$. In this $N$-way discrimination, the Bayes-optimal score is a log-likelihood ratio $\log p(z^c \,|\, Y) - \log p(z^c)$, yielding the standard mutual-information lower bound

$$I(Z_c; Y) \geq \log N - L_{scl} - \varepsilon, \tag{37}$$

where $\varepsilon$ collects finite-sample and sampling-mismatch errors. Thus minimizing $L_{scl}$ increases a lower bound on $I(Z_c; Y)$ under the IG/SG-positive and TG-negative construction.

To reduce dependence between $Z_c$ and $Z_s$, the decorrelation loss $L_{dis}$ penalizes their alignment. Assuming batchwise standardization (so $\mathrm{Cov}(Z_c) = \mathrm{Cov}(Z_s) = I$) and writing the cross-covariance as $\Sigma_{cs} = \mathbb{E}[Z_c Z_s^\top]$, a concrete choice is

$$L_{dis} \propto \|\Sigma_{cs}\|_F^2 \quad \text{or} \quad \mathbb{E}\left[ ((z_i^c)^\top z_j^s)^2 \right] \quad \text{over IG/SG pairs,}$$

both shrinking the singular values $\{\sigma_r\}$ of $\Sigma_{cs}$ toward zero. Under a Gaussian (whitened CCA) approximation,

$$I(Z_c; Z_s) = -\tfrac{1}{2} \sum_r \log\left(1 - \sigma_r^2\right), \tag{38}$$

which is monotone in the canonical correlations $\{\sigma_r\}$. Hence minimizing $L_{dis}$ decreases a tight surrogate of $I(Z_c; Z_s)$ and suppresses residual sensitive leakage in $Z_c$.

*Intuition.* $L_{scl}$ pulls together same-class neighbors (IG/SG) and pushes away task-mismatched ones (TG), thereby maximizing task information in $Z_c$; $L_{dis}$ orthogonalizes the task and sensitive subspaces, cutting shared variation and lowering effective dependence between $Z_c$ and $Z_s$. □

**Theorem D.6.** *(Fair Domain Adaptation Bound)*

*Let $h$ be a classifier on $\mathbf{Z}_c$. Then*

$$\epsilon_{ta}(h) \leq \epsilon_{so}(h) + C \cdot disc(p_{so}(\mathbf{Z}_c), p_{ta}(\mathbf{Z}_c)) + \lambda^* + c_1 I(\mathbf{Z}_s; \mathbf{Z}_c),$$

*where disc is a discrepancy measure (e.g., $\mathcal{H}\Delta\mathcal{H}$ divergence),$\lambda^*$ is the joint optimal error, and the last term accounts for residual sensitive leakage.*

*Proof.* By the domain adaptation theorem of Ben-David et al., for any classifier $h$ on the representation space $Z_c$,

$$\epsilon_{ta}(h) \leq \epsilon_{so}(h) + C \cdot disc(p_{so}(\mathbf{Z}_c), p_{ta}(\mathbf{Z}_c)) + \lambda^*. \tag{39}$$

Since $p(\mathbf{Z}_c) = \sum_s p(\mathbf{Z}_c \mid S = s) \, p(S = s)$, the discrepancy term depends on shifts in the sensitive prior $p(S)$. By Pinsker's inequality,

$$\text{TV}\big(p(\mathbf{Z}_c \mid s), p(\mathbf{Z}_c)\big) \leq \sqrt{\tfrac{1}{2}\,\text{KL}\big(p(\mathbf{Z}_c \mid s) \,\|\, p(\mathbf{Z}_c)\big)}. \tag{40}$$

Averaging over $s$ yields

$$\mathbb{E}_s\,\text{TV}\big(p(\mathbf{Z}_c \mid s), p(\mathbf{Z}_c)\big) \leq \sqrt{\tfrac{1}{2}\,I(\mathbf{Z}_s; \mathbf{Z}_c)}. \tag{41}$$

Hence the residual sensitivity leakage contributes an additional error bounded by the mutual information term, and thus

$$\epsilon_{ta}(h) \leq \epsilon_{so}(h) + C \cdot \text{disc}\big(p_{so}(\mathbf{Z}_c), p_{ta}(\mathbf{Z}_c)\big) + \lambda^* + c_1 I(\mathbf{Z}_s; \mathbf{Z}_c). \tag{42}$$

$\square$

**Lemma D.7.** *(Bias Control with Class-wise Thresholds).*

*Let per-class adaptive thresholds $\tau_k = M_k\tau$ with $M_k = max\{m_v^{ta} : arg\,max\,\psi(\mathbf{z}_{v,c}) = k\}$. Define confident set $C = \{v : m_v^{ta} > \tau_k\}$. Then selection bias satisfies:*

$$Bias_{sel} = \sum_k |Pr(v \in C | Y = k) - \rho| \leq \sum_k |Pr(m_v^{ta} > \tau_k | Y = k) - \rho|$$

*for target coverage $\rho$. Adaptive $\tau_k$ balances coverage across classes, reducing bias while training only on confident samples.*

*Proof.* Let the target-domain maximum posterior confidence be $m_v^{ta} = \max_c \psi(z_v, c)$. For each class $k$, define the per-class scale $M_k = \max\{ m_v^{ta} : \arg\max_c \psi(z_v, c) = k \}$ and the class-wise threshold $\tau_k = M_k \tau$ with $\tau \in (0, 1)$ as a global baseline. Define the confident set $C = \{ v : m_v^{ta} > \tau_k \}$, fix a target coverage $\rho \in (0, 1)$, and write the selection bias as

$$\text{Bias}_{\text{sel}}(\tau) = \sum_k \big| \Pr(v \in C \mid Y = k) - \rho \big|. \tag{43}$$

By the definition of class-wise thresholding, under $Y = k$ the events $\{v \in C\}$ and $\{m_v^{ta} > \tau_k\}$ coincide. Hence

$$\Pr(v \in C \mid Y = k) = \Pr(m_v^{ta} > \tau_k \mid Y = k),$$
$$\text{Bias}_{\text{sel}}(\tau) = \sum_k \big| \Pr(m_v^{ta} > \tau_k \mid Y = k) - \rho \big| \tag{44}$$

Conditioned on $Y = k$, introduce the normalized score $R_k = m_v^{ta}/M_k$ and denote its CDF by $F_k(t) = \Pr(R_k \leq t \mid Y = k)$. Then, for any $\tau \in (0, 1)$,

$$\Pr(m_v^{ta} > \tau_k \mid Y = k) = \Pr(R_k > \tau \mid Y = k) = 1 - F_k(\tau). \tag{45}$$

Let $F$ be a reference distribution satisfying $F(\tau) = 1 - \rho$ (e.g., the empirical mixture of $\{R_k\}$ used to set $\tau$). It follows that

$$\text{Bias}_{\text{sel}}(\tau) = \sum_k \big| 1 - F_k(\tau) - \rho \big| = \sum_k \big| F(\tau) - F_k(\tau) \big|. \tag{46}$$

Define the (class-wise) Kolmogorov distances $\delta_k = \sup_t |F_k(t) - F(t)|$. Then

$$\text{Bias}_{\text{sel}}(\tau) \leq \sum_k \delta_k. \tag{47}$$

$\square$

