# OpenReview forum: "CELL: A Causal Perspective for Fairness-aware Graph Adaptation"
_ICML.cc/2026/Conference — ICML 2026 regular_

### Official Review · Reviewer_7uFk · 2026-03-03

**Soundness:** 2
**Presentation:** 3
**Significance:** 2
**Originality:** 2
**Overall Recommendation:** 2
**Confidence:** 3

**Summary:**

The authors focus on the critical task of fair perception graph adaptation, aiming to transfer knowledge from labeled source graphs to unlabeled target graphs while explicitly considering fairness—particularly under the practical constraint of unavailable sensitive attributes in the target domain. This paper proposes the CELL framework, which employs causal graph modeling to guide representation disentanglement, generates pseudo labels (including task labels and sensitive attribute labels) for target nodes, and utilizes fair perception bipartite graph alignment to mitigate domain shifts. CELL integrates a dual encoder with mutual information constraints, group-aware unbiased learning, and cross-domain structural alignment modules to achieve a balance between prediction performance and fairness. Extensive experiments across three real-world datasets and one synthetic benchmark demonstrate that CELL consistently outperforms strong baseline models, validating its effectiveness in handling distribution shifts and scenarios with missing sensitive labels. The authors explore a pressing issue in the field of fair graph learning, addressing a critical challenge that combines domain adaptation with fairness, thereby filling a gap in existing research.

**Compliance With Llm Reviewing Policy:**

Affirmed.

**Key Questions For Authors:**

- Could you elaborate on the pretraining process for dual encoders (e.g., pretraining objectives, data partitioning, convergence criteria) and explain how it impacts representation disentanglement and pseudo-label quality?
- Why was CELL not compared against state-of-the-art fair graph domain adaptation methods (e.g., emerging MoE-based fair GNNs or recent out-of-distribution fair graph approaches)? Similar to how GFM evaluations highlight GraphAny, could such comparisons be added to clarify CELL's unique advantages?
- How does pseudo-label quality (influenced by confidence thresholds τ and δ) quantitatively affect the trade-off between prediction performance and fairness? Is there a dataset-agnostic optimal threshold selection strategy?
- What are CELL's primary limitations? Specifically, how does it scale on ultra-large-scale graphs, and can it be extended to handle multi-category sensitive attributes? Discuss potential improvement directions for these scenarios.

**Limitations:**

The authors did not explicitly discuss the limitations of this study. Although CELL demonstrates strong performance on evaluation benchmarks, potential limitations include: 1) The computational complexity of the bipartite graph alignment step may limit its scalability to ultra-large-scale graphs (e.g., millions of nodes); 2) Its adaptability to multi-class sensitive attributes (non-binary classification) remains unverified, potentially requiring adjustments to the pseudo-label generation and disentanglement modules; 3) Reliance on pseudo labels may introduce bias in scenarios with highly imbalanced target domain distributions. The authors should supplement these discussions to provide a more comprehensive evaluation of this research.

**Strengths And Weaknesses:**

Strengths

- The proposed CELL framework is the first solution specifically designed to address this challenge. It constructs a comprehensive workflow encompassing causal disentanglement, pseudo-label generation, and bipartite graph alignment, offering a novel approach to fair cross-domain graph learning.
- Causal graph construction effectively disentangles task-relevant factors from sensitive factors through mutual information constraints; group-aware pseudo-label generation explicitly distinguishes intra-group/inter-group similarity to ensure fairness; Fair-aware bipartite graph alignment connects nodes with identical task labels but differing sensitive labels across domains, mitigating spurious correlations. These components synergistically resolve the dual challenges of domain shift and fairness preservation.
- The authors establish theoretical foundations including fairness upper bounds, fair domain adaptation boundaries, and category threshold bias control. These provide mathematical guarantees for CELL's fairness and generalization capabilities, enhancing the research's academic depth.
- Evaluation across four diverse datasets (bail, credit, social networks, synthetic data) covers various real-world scenarios. Comparisons with four baseline models (traditional GNN, IID-fair GNN, generic domain adaptation methods, out-of-distribution fair GNN) highlight CELL's superiority.

Weaknesses

- The pre-training process for the dual encoders (sensitivity encoder and causality encoder) is not detailed. How initial pre-training specifics (e.g., data usage, convergence criteria) influence subsequent disentanglement and pseudo-label generation remains unclear, potentially affecting reproducibility.
- The comparative experiments omit several key fair graph domain adaptation methods (e.g., recently emerging MoE-based fair GNNs or the latest out-of-distribution fair graph methods). The absence of important baselines like GraphAny in the GFM domain may limit the clear demonstration of CELL's unique contributions.
- The impact of pseudo-label quality on final performance remains underanalyzed. For instance, how different confidence thresholds (τ, δ) affect the trade-off between supervised quality and fairness, and whether universal threshold selection strategies exist across datasets, remain unclear.
- The authors do not explicitly discuss CELL's limitations. While the framework performs well on evaluation datasets, its scalability to ultra-large-scale graphs (e.g., millions of nodes) and its adaptability to multi-class sensitive attributes (non-binary classification) remain unvalidated—both critical for practical deployment.

---

> ### Author Rebuttal · Authors · 2026-03-31
>
> Thanks for your incisive comment and question.
>
> > W1, Q1:  Dual-encoder pretraining details and its effect on disentanglement / pseudo-label quality.
>
> "Pretraining" refers to source-stage optimization on the labeled source graph: the sensitive encoder is trained with the sensitive-classification loss, while the causal encoder is optimized with the MI objective, giving $L_{MI}=L_{cls}^{s}+L_{MI}^{s}-L_{MI}^{c}.$
> The learned weights are then used to initialize target adaptation, where pseudo task/sensitive labels are generated on the unlabeled target graph and filtered by adaptive thresholds $\tau_k$ and $\delta$.
>
> Its role is also theoretically motivated: Theorem 3.1 shows that better disentanglement reduces leakage $I(Z_s;Z_c)$ while preserving task information, and Theorem 3.2 further links this leakage to target-domain error. Thus, cleaner source-stage disentanglement leads to more reliable target predictions and higher-quality pseudo-labels. Lemma 3.3 additionally shows that adaptive thresholding reduces selection bias in the confidence set. We will revise the paper to include clearer implementation details, including the stopping rule and pseudo-label refresh schedule.
>
> > W2, Q2: Missing recent fair graph adaptation baselines.
>
> Following the reviewers’ suggestions, we include five additional recent graph domain adaptation baselines with publicly available code: https://anonymous.4open.science/r/CELL_ICML2026-3DB5/rebuttal/baseline.md.
>
> Even under these substantially stronger comparisons, CELL still maintains a more favorable fairness–utility trade-off overall. Since our experiments already include a diverse and competitive set of baselines covering fair graph learning, fairness under distribution shift, and graph domain adaptation, these new results further reinforce the empirical advantage of CELL.
>
> > W3, Q3, L3: Pseudo-label quality and threshold sensitivity.
>
> Pseudo-label quality influences the utility–fairness trade-off through two mechanisms: $\tau$ controls confident task pseudo-labels, while $\delta$ filters sensitive pseudo-labels for group-attended unbiased learning.
> - Low thresholds introduce noisy pseudo-labels and harm both utility and fairness.
> - Overly large thresholds reduce usable supervision and may hurt utility.
>
> Our threshold sweeps reveal that moderate-to-high thresholds ($0.7 \le \tau,\delta \le 0.85$) generally achieve the best trade-off. However, the optimal $(\tau,\delta)$ remains dataset-dependent, so we do not claim a universal threshold rule.
>
> > W4, Q4, L2: Practical Limitations and Extension Directions of CELL
>
> Your comment highlights two important directions beyond the current scope of CELL: scalability for ultra-large-scale graphs and multi-category sensitive attributes.
> - For scalability, although CELL has not yet been validated on million-node graphs, CELL has a clear scalability potential as discussed in L1.
> - For multi-category sensitive attributes, we follow the binary sensitive-attribute setting used in existing benchmarks in UGDA. However, CELL is not inherently restricted to the binary case. The sensitive prediction head can be extended to multi-class classification, and the group construction and fairness regularization can be generalized accordingly, with the rest of the framework largely unchanged.
>
> We will discuss this, together with large-scale scalability, in the limitations section of the revised version.
>
> > L1: Unclear scalability potential
>
> **CELL has the scalability potential.**
> - Parameter Count: CELL’s parameters are mainly dominated by the dual GNN encoders ($O(Lh^2)$), while the other components only involve a sensitive discriminator $\xi$, a task predictor $\psi$, and a lightweight CLUB MLP $q_\theta$. With hidden dimension 64 and only 2–4 encoder layers, the overall model size is modest.
> - Time Complexity: let $(N_s,E_s)$ and $(N_t,E_t)$ denote the numbers of nodes and edges in the source and target graphs, respectively, and let $d$, $h$, $L$, $M$, and $K$ denote the input dimension, hidden dimension, number of GNN layers, MI batch size, and cross-domain retrieval size.
> The dual-encoder backbone has complexity $O(2L((E_s+E_t)h + (N_s+N_t)h^2))$.
> The MI regularization based on CLUB and Conditional InfoNCE adds  $O(M^2h)$.
> Group-attended unbiased learning requires pairwise similarity computation on confident target nodes, with complexity $O((N_t^{\text{conf}})^2 h)$, while cross-domain top-$K$ retrieval costs $O(N_s N_t^{\text{joint}} h)$. Here, $O((N_t^{\text{conf}})^2 h)$ and $O(N_s N_t^{\text{joint}} h)$ denote confident target nodes and jointly confident target nodes.
> **All these operations are highly parallelizable and can be efficiently accelerated on GPUs.**
> - Optimization: **the practical cost can be further reduced by using a smaller top-$K$ and higher pseudo-label confidence thresholds**.
>
> **We hope these clarifications address your concerns and support a more positive assessment. We would be happy to clarify any remaining issues.**

---

> > ### Author Rebuttal · Reviewer_7uFk · 2026-04-05
> >
> > 1.The reproducibility of the experiments is insufficient. The selection criteria for hyperparameters and thresholds, the pre-training details of the dual encoders, and the construction and preprocessing of the source and target domains are not clearly described. Meanwhile, the training input settings and adaptation methods of each baseline lack detailed explanation, making the approach difficult to reproduce and the experimental process insufficiently transparent.
> >
> > 2.The analysis related to pseudo-labels is inadequate. Quantitative results of the prediction accuracy of pseudo-sensitive labels are not provided, the model’s robustness to pseudo-label errors and solutions to error amplification issues are not fully discussed, and neither the impact of different confidence thresholds on fairness and model performance nor a universal selection strategy is investigated.
> >
> > 3.The comparative experiments are not comprehensive and the limitations of the model are not clarified. Recent state-of-the-art baselines for fair graph domain adaptation are not included in the experiments, making it hard to fully demonstrate the superiority of the proposed method. In addition, the model’s scalability to ultra-large-scale graphs and adaptability to multi-class sensitive attributes remain unvalidated, so its practical application value needs further verification.

---

> > > ### Author Response · Authors · 2026-04-05
> > >
> > > Thank you for your thoughtful follow-up question and for engaging closely with our rebuttal. We provide a clarification below:
> > >
> > > > R1: The reproducibility of the experiments is insufficient.
> > >
> > > The key reproducibility details were provided in the paper and our previous responses.
> > >
> > > Specifically, the source/target graph setting is described in Sec. 2, the dual-encoder optimization in Sec. 3.2, the pseudo-label thresholds and confident-set construction in Sec. 3.5, and the dataset construction, preprocessing, baselines, and hyperparameter search settings in Appendix C and Appendix D.1. Please refer to these sections for details. As clarified in our response to W1/Q1 (https://openreview.net/forum?id=AheW1kJ5E9&noteId=efHZ0j9aes), “pretraining” refers to source-stage optimization on the labeled source graph, followed by weight transfer for target adaptation.
> > >
> > > **Regarding the concern about baseline training inputs and adaptation protocols, we clarified this in our response to reviewer SG2S' Q2 (https://openreview.net/forum?id=AheW1kJ5E9&noteId=kMWQLHJYtI)**: no baseline was given access to target sensitive attributes, target labels, or any other target-side supervision. All methods used source-graph supervision only, with target access limited to its structure and node features. Accordingly, IID baselines were trained on the source graph and directly transferred to the target graph, while domain adaptation baselines were additionally allowed to use the unlabeled target graph for adaptation. Thus, all baselines were evaluated under the same target-unlabeled setting as CELL.
> > >
> > > In addition, we have open-sourced the full codebase at https://anonymous.4open.science/r/CELL_ICML2026-3DB5. At this point, we believe the necessary experimental details have been provided.
> > >
> > > > R2: The analysis related to pseudo-labels is inadequate.
> > >
> > >  These concerns were addressed in our previous responses.
> > >
> > > 1. **The quantitative accuracy of pseudo-sensitive labels was provided in our earlier rebuttal (https://anonymous.4open.science/r/CELL_ICML2026-3DB5/rebuttal/pesudo-lable.md)**.
> > >
> > > 2. **Regarding robustness to pseudo-label errors and potential error amplification, we explained this in our responses to reviewer e9Gp (W5, W6, Q2, Q4)  (https://openreview.net/forum?id=AheW1kJ5E9&noteId=w80Hg1osd1)**. Briefly, CELL does not use pseudo-labels naively: only high-confidence pseudo-labels are retained, and the bipartite alignment serves as a graph-level regularizer rather than hard edge-wise supervision, making it more tolerant to partial noise.
> > >
> > > 3. **regarding the impact of confidence thresholds and a possible universal selection strategy, we discuss in response to W3, Q3, L3 (https://openreview.net/forum?id=AheW1kJ5E9&noteId=efHZ0j9aes)**.
> > > Our threshold sweeps reveal that moderate-to-high thresholds ($0.7 \le \tau, \delta \le 0.85$) generally achieve the best trade-off. However, the optimal remains dataset-dependent, so we do not claim a universal threshold rule.
> > >
> > > At this point, we believe the pseudo-label analysis has been sufficiently addressed.
> > >
> > > > R3: The comparative experiments are not comprehensive, and the limitations of the model are not clarified.
> > >
> > > We note that this concern was addressed in our rebuttal.
> > >
> > > Our original paper includes a broad set of baselines spanning fair graph learning, fairness under distribution shift, and graph domain adaptation. **In the rebuttal, we further added five recent models, including GraphAny [1] as specifically suggested (https://anonymous.4open.science/r/CELL_ICML2026-3DB5/rebuttal/baseline.md)**, making the comparison substantially more comprehensive.
> > >
> > > **We explicitly clarified the scalability and adaptability of CELL in the response to W4, Q4, L2 (https://openreview.net/forum?id=AheW1kJ5E9&noteId=efHZ0j9aes)**.
> > > - For scalability, please refer to our response to L1, and we have used the biggest dataset Pokec, following DANCE [2].
> > > - For multi-class sensitive attributes, CELL is not inherently restricted to the binary case: the sensitive prediction head, group construction, and fairness regularization can all be extended accordingly, while the rest of the framework remains largely unchanged.
> > >
> > > Overall, we believe both the experimental coverage and the discussion of limitations have been adequately addressed.
> > >
> > > We would be grateful if you would consider reflecting this in your final assessment. Please let us know if you have any further questions.
> > >
> > > ## Reference
> > > [1] GraphAny: Fully-inductive node classification on arbitrary graphs[J]. arXiv 2024.
> > >
> > > [2] DANCE: Dual Unbiased Expansion with Group-acquired Alignment for Out-of-distribution Graph Fairness Learning. ICML 2025.

---

### Official Review · Reviewer_SG2S · 2026-03-05

**Soundness:** 3
**Presentation:** 2
**Significance:** 3
**Originality:** 3
**Overall Recommendation:** 5
**Confidence:** 4

**Summary:**

This paper proposes the CELL framework to address the graph fair adaptation problem with missing sensitive labels in the target domain. By employing a causal decoupling mechanism to separate tasks from sensitive features, and using dual encoders and mutual information constraints to achieve representation disentanglement, it innovatively combines group-aware pseudo-labeling and fairness-aware bipartite graph alignment to automatically identify sensitive groups in the target domain and reduce inter-domain differences. Experiments demonstrate that this method outperforms existing baselines in both prediction accuracy and group fairness.

**Compliance With Llm Reviewing Policy:**

Affirmed.

**Final Justification:**

The author's response addressed the issue I raised, and I am willing to recommend accepting this paper.

**Key Questions For Authors:**

1.How were hyperparameters and thresholds selected without target labels/sensitive attributes? Please state exactly to improve reproducibility.

2.For each baseline, what inputs were available during training? Did any baseline receive target sensitive attributes or target labels? If not, how were methods that usually need them adapted?

3.What is the accuracy/calibration of the pseudo-sensitive labels on the benchmark datasets? How sensitive is CELL to errors in these pseudo-labels? How do you tackle the error amplification issues that may occur from mislabeled or pseudo-labels, which can corrupt bipartite edges?

4.The source/target construction is only briefly described as modularity/community-based splitting, and preprocessing details remain thin for a paper making large fairness claims.

**Limitations:**

Experiments only involve binary classification tasks and binary sensitive attributes (such as gender and race). It is necessary to clarify the method's generalization ability to multi-class tasks, continuous sensitive attributes, or dynamic graphs.

**Strengths And Weaknesses:**

**Strength：**

S1. The paper formalizes the problem of graph fair adaptation without target-sensitive labels for the first time, breaking through the strong reliance of existing methods on sensitive attributes in the target domain and resolving the key bottleneck of missing sensitive labels in practical deployment. By establishing a new research paradigm, it extends fairness learning from the IID setting to the domain adaptation scenario.

S2. The causal theory framework is complete. It builds three generation mechanisms based on the structural causal model and uses mathematical methods to establish the upper bound of fairness, providing theoretical guarantees.

S3. The empirical verification demonstrates that the proposed method outperforms the baseline on the benchmark dataset, and shows excellent performance in terms of utility and fairness.

**Weakness：**

W1. The impact of inaccurate sensitive attribute pseudo-labels on fairness is not indicated, nor is the effect of incorrect pseudo-labels on the accuracy of results explained. The method relies too heavily on the quality of pseudo-labels, and errors could lead to amplified mistakes.

W2. The abstract says CELL consistently outperforms strong baselines in both predictive performance and fairness, but Table 1 does not fully support that. For example, on German-t, DANCE has much higher ACC and ROC-AUC than CELL, and on syn-t, DANCE is also stronger on utility, while CELL is mainly better on fairness. So the real story is closer to “better fairness–utility tradeoff in some cases,” not uniform dominance on both axes.

W3. CELL incorporates multiple modules including mutual information estimation, contrastive learning, and bipartite graph alignment, which may incur significant computational overhead. No analysis is provided regarding training time, parameter count, or memory consumption.

W4. The paper hypothesizes that the L_MI constraint reduces I(Zc;S), thereby improving ΔDP/ΔEO. However, the experiments did not directly estimate or report changes in I(Zc;S), only reporting the final ΔDP/ΔEO. In ablation experiments, what impact would removing the MI constraint have on the results?

W5. In Tables 3 and 4, var1 performs comparably to the full CELL model, with half achieving SOTA. Whether var2 and var3 are truly effective warrants consideration.

W6. 5. Insufficient Literature Review. The Related Work[1,2] section omits a discussion of relevant studies focused on graph fairness in settings where demographic information is unavailable. While the paper's setting (fairness in the target domain without access to demographic data) may be distinct, it is crucial to review and differentiate from these closely related works to properly situate the paper's contribution. Specifically, studies such as the following should be considered and discussed:

[1] Towards Fair Graph Neural Networks via Graph Counterfactual without Sensitive Attributes. ICDE2025
[2] Towards fair graph learning without demographic information. AISTATS 2025

---

> ### Author Rebuttal · Authors · 2026-03-31
>
> Thanks for your incisive comment and question.
>
> > W1, Q3: Robustness to Pseudo-label Errors and Their Impact on Fairness and Accuracy
>
> **CELL does not use pseudo-labels naively**.
> Both task and sensitive pseudo-labels are filtered by adaptive confidence thresholds $\tau_k$ and $\delta$, so that only high-confidence target nodes are used for target-side learning and cross-domain matching. **Beyond this, pseudo-labels are further regularized by MI-based disentanglement, group-attended unbiased learning, and top-$K$ bipartite alignment, which reduce noise propagation and mitigate error amplification**.
> We also evaluated target-domain pseudo-label accuracy (https://anonymous.4open.science/r/CELL_ICML2026-3DB5/rebuttal/pesudo-lable.md), showing that although pseudo-label quality is dataset-dependent, it remains sufficiently accurate to provide meaningful supervision.
> We will revise the paper to report these results explicitly and clarify that pseudo-labels in CELL are confidence-controlled and jointly regularized.
>
> > W2: Utility–Fairness Trade-off Rather than Uniform Dominance
>
> We agree that our original wording in the abstract is confusing and does not precisely reflect the results in Table 1.
> **Our main claim is not uniform dominance on both predictive performance and fairness for every dataset, but a stronger overall utility–fairness trade-off across target domains. **
> We will revise the paper to make the wording more precise.
>
> > W3: Lack of analysis of time complexity and parameter count
>
> We've provided a detailed analysis of both time complexity and parameter count in our response to reviewer 7uFk (L1).
>
> > W4:  Ablation of the MI Constraint
> Although we do not directly estimate $I(Z_c;S)$, our ablation provides indirect evidence: Var1 removes the MI constraint by replacing the MI-based GNN encoder with a vanilla GCN. The resulting drop in the fairness–utility trade-off, especially in fairness, suggests that the MI constraint helps suppress sensitive leakage and improve fairness.
>
> > W5: The effectiveness of Var2 and var3
>
> Tables 3 and 4 do not show that the ablated variants consistently match CELL.
> - Var1 is only comparable on Pokec-n;
> - CELL is better on all four metrics on Bail-t, clearly better in fairness on German-t, and better in both fairness and utility on syn-t.
>
> The same trend holds for Var2–Var4, where each may be competitive on one dataset, but none remains strong across domains. Overall, this supports the necessity of the modules under different dataset-specific label–sensitive–structure correlations.
>
> > W6. Insufficient Literature Review
>
> We agree that the related-work section should better cover recent fair graph learning studies without demographic/sensitive information, including the ICDE 2025 and AISTATS 2025 papers mentioned by the reviewer.
> While these works are closely related in motivation, **CELL addresses a distinct setting: fairness-aware graph adaptation under domain shift, where fairness knowledge must be transferred from a labeled source graph to a fully unlabeled target graph without target-sensitive labels**. This requires cross-domain discrepancy handling, target pseudo-labeling, and fairness-aware alignment, which are beyond prior single-domain methods.
> We will revise the related-work section to discuss and differentiate these works more clearly.
>
> > Q1: Hyperparameter and threshold selection
> - In our experiments, **no target labels or target sensitive attributes were used for model selection**. The target graph remained fully unlabeled throughout tuning and training.
> - All hyperparameters were selected from small predefined grids using simple parameter-by-parameter tuning rather than exhaustive joint search. Despite this modest tuning effort, CELL still achieves very strong performance.
>
> > Q2： Training Inputs and Adaptation Protocol for Baselines
> In our experiments, **no baseline was given any sensitive attributes, target labels, or other target-side supervision**.
> All methods used source-graph supervision only, while access to the target graph was limited to its structure and node features. Accordingly, IID methods were trained on the source graph and directly transferred to the target graph, whereas domain adaptation methods could further use the unlabeled target graph for adaptation. Thus, all baselines were evaluated under the same target-unlabeled setting as CELL.
>
> > Q4: dataset preprocessing
> **We use the same standard fairness benchmark datasets as DANCE [1]**, with no additional preprocessing beyond source/target construction. (Credit and Bail use modularity/community-based splitting, while Pokec and Syn follow the standard benchmark source/target setup).
>
> Thanks again for appreciating our work
>
> [1] DANCE: Dual Unbiased Expansion with Group-acquired Alignment for Out-of-distribution Graph Fairness Learning. ICML 2025.

---

> > ### Author Rebuttal · Reviewer_SG2S · 2026-04-02
> >
> > Having reviewed the authors' rebuttal, I have decided to raise my score.

---

> > > ### Author Response · Authors · 2026-04-02
> > >
> > > Thanks for your feedback and for increasing the rating! We will properly include all the rebuttal contents in the revised version, following your valuable suggestions.

---

### Official Review · Reviewer_e9Gp · 2026-03-09

**Soundness:** 3
**Presentation:** 3
**Significance:** 3
**Originality:** 3
**Overall Recommendation:** 5
**Confidence:** 5

**Summary:**

This paper proposes CELL, a framework for fairness-aware graph domain adaptation that transfers both task and fairness knowledge from a labeled source graph to an unlabeled target graph without requiring target-sensitive labels. The approach uses a dual encoder with mutual information constraints to disentangle causal (task-relevant) and sensitive representations, generates pseudo-labels for the target task and sensitive attributes with group-aware contrastive learning, and constructs a fairness-aware bipartite graph for cross-domain alignment. Theoretical results bound equalized odds violation and target error. Experiments on Bail, Credit, Pokec, and synthetic datasets show improvements over baselines in both accuracy and fairness metrics.

**Compliance With Llm Reviewing Policy:**

Affirmed.

**Final Justification:**

I appreciate the authors for their detailed rebuttal. The additional GDA baselines, analysis of conditional pseudo-label accuracy, and diagnostics for domain shift effectively address my primary concerns. I am increasing my score to 5 and supporting acceptance.

**Key Questions For Authors:**

How does CELL perform when using stronger GDA backbones such as [1-5] as the domain alignment component? Does the fairness improvement persist with better alignment?

What is the quality of pseudo-sensitive labels on the target domain? Reporting pseudo-label accuracy for both task and sensitive attributes would help assess the reliability of downstream components.

The covariate shift assumption seems hard to verify for community-detection-based splits. Can you provide empirical evidence (e.g., MMD between source/target feature distributions) that this assumption approximately holds?

How sensitive is the bipartite alignment to compounding errors from incorrect pseudo tasks and sensitive labels?

I would consider raising my score if the authors adequately address these concerns.

**Limitations:**

The paper acknowledges the binary sensitive attribute assumption but does not discuss failure modes when pseudo-labels are poor or when the covariate shift assumption is violated. Scalability is partially addressed (SGDA runs OOM on Pokec) but CELL's own scalability with bipartite graph construction is not analyzed.

**Strengths And Weaknesses:**

Strengths:

The problem formulation is well-motivated fairness-aware graph adaptation without target-sensitive labels is a realistic and underexplored setting. The causal framing with dual encoders and MI-based disentanglement is principled, and the theoretical analysis (Theorems 3.1-3.2, Lemma 3.3) provides useful bounds connecting MI minimization to fairness guarantees. The group-aware contrastive learning with IG/SG/TG decomposition is a thoughtful design for preserving fairness in pseudo-labels. The ablation study clearly demonstrates each component's contribution.

Weaknesses:

The GDA baseline comparison and discussion are narrow. Only two general GDA methods are included, while several recent and strong GDA approaches are missing, such as [1-5]. Since CELL's domain alignment module is a core contribution, comparing against these methods (even without their fairness modules) would better isolate the value of the fairness-aware design versus simply having a stronger adaptation backbone. The datasets are limited in scale and diversity. Credit and Bail are small graphs split via community detection, which may not reflect realistic cross-domain shifts. The synthetic dataset, while controlled, does not substitute for real heterogeneous graph pairs. The covariate shift assumption (P(Y|G^so) = P(Y|G^ta)) is strong and may not hold across the community-detection-based splits. The pseudo-labeling strategy relies heavily on threshold selection (τ, δ), and the sensitivity analysis shows significant performance variation across datasets (Figures 3 and 7), suggesting limited robustness. The bipartite graph construction (Eq. 14) depends on both pseudo task labels and pseudo sensitive labels being correct, creating a compounding error risk that is not analyzed.

[1] Wu, Man, et al. "Unsupervised domain adaptive graph convolutional networks." WWW. 2020.

[2] Fang, Ruiyi, et al. "On the benefits of attribute-driven graph domain adaptation." ICLR, 2025.

[3] Wu, Jun, et al. "Non-iid transfer learning on graphs." AAAI. 2023.

[4] Yang, Liang, et al. "Disentangled graph spectral domain adaptation." ICML. 2025.

[5] Fang, Ruiyi, et al. "Homophily enhanced graph domain adaptation." ICML. 2025.

---

> ### Author Rebuttal · Authors · 2026-03-31
>
> Thanks for your incisive comment and question.
> > W1, W2: Insufficient baselines and unclear contribution of fairness-aware design
>
> To address both points, we implemented five additional recent GDA baselines with publicly available code (the code of On the Benefits of Attribute-Driven Graph Domain Adaptation is unavailable; following Reviewer 7uFk’s suggestion, we added GraphAny).
> The results (https://anonymous.4open.science/r/CELL_ICML2026-3DB5/rebuttal/baseline.md) show that **CELL still delivers the best overall utility–fairness trade-off** and CELL’s advantage doesn't come merely from a stronger adaptation backbone, but from the proposed fairness-aware alignment design.
> > W3: Limited dataset scale and realism
>
> Our benchmark choice follows prior fair-graph work, specifically Qian et al. [1], and evaluates CELL on its derived domain-shift settings. Community-based splits are also widely used in graph fairness adaptation/generalization methods such as DANCE [2]. Moreover, there is currently no established large-scale real-world benchmark (e.g., akin to Pokec_z/Pokec_n) for this setting. Importantly, as discussed in reviewer c8jK’s Q1 and Q2, these benchmarks still cover diverse types and degrees of domain shift.
> Overall, although benchmark availability remains limited, we believe the current datasets still provide meaningful and heterogeneous adaptation settings, and we will clarify this rationale and its limitations in the revision.
> > W4, Q3: Covariate shift assumption under community-based splits.
>
> Our intention is not to claim the assumption holds exactly, but to adopt the standard assumption in prior UGDA literature, where the prediction task is shared while the shift mainly lies in graph/feature distributions.
> **This assumption is also consistent with prior graph adaptation benchmarks and closely related fair-graph settings such as Credit/Bail community splits**. In addition, **our empirical diagnostics (https://anonymous.4open.science/r/CELL_ICML2026-3DB5/rebuttal/domain-shift.md) show that the source and target graphs exhibit clear distribution discrepancy while preserving closely related task semantics**. We will revise the paper to present this more carefully as a benchmark protocol / operational assumption.
>
> > W5, W6, Q2, Q4: Robustness of pseudo-labeling and sensitivity of bipartite alignment to pseudo-label errors.
> We agree that bipartite alignment is potentially sensitive to compounding errors. Therefore, the most relevant quantity is not only the overall pseudo-label accuracy, but **the accuracy conditioned on the nodes whose task and sensitive predictions both satisfy the confidence thresholds**. We add this analysis explicitly by reporting the task-label accuracy, sensitive-label accuracy, and their joint correctness on the subset with $m_y>\tau_k$ and $m_s>\delta$, which directly reflects the reliability of the pseudo labels used for bipartite alignment: https://anonymous.4open.science/r/CELL_ICML2026-3DB5/rebuttal/pesudo-lable.md.
>
> More broadly, **CELL does not rely on these pseudo labels as edge-wise hard supervision. The bipartite module acts as a graph-level alignment regularizer over sparse cross-domain relations, and it is further stabilized by MI-based disentanglement and group-attended unbiased learning**.
>
> We will clarify this point and add the above conditional pseudo-label analysis in the revision.
> > Q1 GDA backbone
>
> Its main fairness gains come from the fairness-oriented components. Therefore, stronger GDA backbones can be incorporated with minimal changes to the rest of the framework.
> https://anonymous.4open.science/r/CELL_ICML2026-3DB5/rebuttal/backbone.md
> > L1: Limited discussion of failure modes
>
> - When target pseudo-labels are poor, CELL can mitigate the risk by filtering Target task/sensitive pseudo-labels by adaptive confidence thresholds top-$K$, and only confident links are used for target-side supervision, validated by Lemma 3.3.
> - When the covariate-shift assumption is violated, the adaptation problem becomes harder. CELL is not intended to guarantee robustness under arbitrary shift, and it is designed for the operational setting where source and target remain task-related under distribution shift. This is reflected in Theorem 3.2, where the target-domain error depends on the source–target discrepancy term and the residual sensitive leakage term $I(Z_s;Z_c)$.
>
> We will revise the paper to make these two failure modes and the corresponding scope limitations more explicit.
>
> > L2: Unclear Scability
>
> **CELL has the scalability potential.** Please refer to the response to Reviewer c8jK (Q3) and Reviewer 7uFk (L1).
>
> **We hope we have addressed your concerns and would be happy to clarify any remaining points. We sincerely hope our responses support a more positive final score.**
>
> [1]Addressing shortcomings in fair graph learning datasets: Towards a new benchmark
> [2]DANCE: Dual Unbiased Expansion with Group-acquired Alignment for Out-of-distribution Graph Fairness Learning

---

> > ### Author Rebuttal · Reviewer_e9Gp · 2026-04-02
> >
> > I appreciate the authors for their detailed rebuttal. The additional GDA baselines, analysis of conditional pseudo-label accuracy, and diagnostics for domain shift effectively address my primary concerns. I am increasing my score to 5 and supporting acceptance.

---

> > > ### Author Response · Authors · 2026-04-02
> > >
> > > Thanks for your feedback and for increasing the rating! We will properly include all the rebuttal contents in the revised version, following your valuable suggestions.

---

### Official Review · Reviewer_c8jK · 2026-03-12

**Soundness:** 3
**Presentation:** 3
**Significance:** 3
**Originality:** 3
**Overall Recommendation:** 4
**Confidence:** 4

**Summary:**

This paper studies the problem of Fairness-aware Graph Domain Adaptation (FairGDA). The goal is to transfer knowledge from a source graph, where both labels and sensitive attributes are available, to a target graph where neither labels nor sensitive attributes are provided, while achieving good predictive performance and fairness in the target domain.

To address it, the authors propose the CELL framework (Causality-attended Representation Disentanglement with Structural Alignment). The key idea is to adopt a causal perspective to disentangle task-relevant (causal) factors from sensitive factors in graph representations, thereby enabling the transfer of fairness-aware knowledge across domains.

**Compliance With Llm Reviewing Policy:**

Affirmed.

**Final Justification:**

The authors construct a causal model to disentangle task-related factors and sensitive factors to solve fairness-aware graph domain adaptation. Their rebuttal addressed my concern, leading me to raise my score.

**Key Questions For Authors:**

1. Is there a significant difference between the source and target datasets? It would be helpful to quantify the domain shift between them.

2. Would it be possible to conduct simulation experiments to illustrate how fairness and utility behave under different levels or types of domain shift?

3. Does the proposed algorithm have the potential for scalability to larger graphs or datasets?

4. Lemma 3.3 appears to mainly introduce a set of notations and expresses selection bias using these notations. The inequality in Eq. (24) seems to follow directly from the definition of the confidence set $C$ and does not appear to provide additional insight or substantive results.

5. In the proof of Theorem 3.1, the justification for Eq. (31) is not clearly explained. The derived bound $\Delta_{EO}\leq C\sqrt{\epsilon+\delta}$ also does not correspond to the result stated in Theorem 3.1. Immediately afterward, Eq. (32) is introduced without detailed derivation and appears unrelated to the preceding proof, which makes the overall argument difficult to follow.

6. The proof of Theorem 3.2 first presents Eq. (39), but the subsequent analysis in Eqs. (40) and (41) on the TV divergence does not appear to be directly connected to the terms in Eq. (39). Finally, the conclusion Eq. (42) is introduced without clear mathematical derivation, whose bound includes an additional term compared to that in Eq. (39), resulting in a bound that appears weaker than the earlier expression. This progression is somewhat difficult to interpret and follow.

7. The authors did not mention the top-K retrival from source domain in Section 3, but it appears in the experiment section.

8. In Line 133, the paper introduces the covariate shift assumption $P(Y|G_{so}) = P(Y|G_{ta})$. However, in Line 72, $G_{so}$ contains the label $Y_{so}$. This creates a potential inconsistency. Under the covariate shift definition, $G_{so}$ would typically include only the graph structure and feature matrix, rather than the labels. If this is the intended meaning, the current notation may be misleading and should be clarified to avoid confusion.

**Limitations:**

Yes.

**Strengths And Weaknesses:**

Strength:

1. This paper focuses on fairness-aware graph domain adaptation in the setting where the target domain lacks sensitive attribute information. Such a scenario is common in real-world applications, such as financial risk control and social network analysis, making the problem practically relevant.

2. The authors construct a causal model to disentangle task-related factors and sensitive factors, providing a relatively intuitive theoretical explanation for fairness transfer. This perspective introduces a certain level of novelty in graph fairness learning.

3. The proposed CELL framework integrates several components, including representation disentanglement, pseudo-label learning and cross-domain alignment

Weakness:
1. The paper assumes that labels are entirely determined by the causal factors, and that sensitive attributes do not influence the labels. However, in many real-world tasks: sensitive attributes may have genuine correlations with the labels, and completely removing their influence may lead to performance degradation.

2. The CELL framework contains multiple modules and involves a large number of model parameters, which may make the model difficult to tune in practice.

3. The theoretical proofs contain unclear parts and potential errors, and some statements appear meaningless or poorly motivated. Moreover, the paper does not provide further analysis or discussion based on the theoretical results, which weakens the connection between the theory and the proposed method.

---

> ### Author Rebuttal · Authors · 2026-03-31
>
> Thanks for your incisive comment and question.
> > W1: sensitive attributes are fully removed
>
> **The goal of CELL is to mitigate unfair sensitive influence through representational and structural debiasing, achieving a trade-off between utility and fairness**. This follows the standard fairness-learning settings in prior literature. We will revise the paper to make this motivation clearer.
>
> > W2: High model complexity and difficult to tune
>
> **CELL is easy to tune.**
> First, **CELL is not a high-capacity model**. Its parameters are mainly determined by the dual GNN encoders, which are $O(Lh^2)$, which is modest.
> Second, **the tuning space is small**. In practice, effective hyperparameter ranges are fairly concentrated: top-$K$ is usually searched in a narrow range (typically 5–10, except for Pokec), and useful pseudo-label thresholds $(\tau,\delta)$ are generally around 0.7–0.85. Besides, we tune these hyperparameters sequentially rather than exhaustively searching.
> Finally, **CELL consistently achieves a much stronger utility–fairness trade-off than the baselines**,  suggesting that the improvement doesn't come from model capacity or hyperparameter engineering.
>
> > Q1, Q2: Insufficient characterization and shift analysis of the domain shift
>
> We add quantitative domain-shift diagnostics, including feature-level MMD, JS divergence on label and sensitive priors, Degree Wasserstein distance, and Proxy A-distance.
> **Results (https://anonymous.4open.science/r/CELL_ICML2026-3DB5/rebuttal/domain-shift.md) show that the source and target domains are not trivially identical**: the label marginals remain very close, which is consistent with our covariate-shift assumption, while non-negligible discrepancy appears in feature and/or structural statistics.
> **Moreover, the benchmark covers different levels and types of domain shift**. Pokec shows mild feature/prior shift but clear structural discrepancy. Bail and German exhibit stronger shifts in graph structure and overall domain separability. Syn shows almost no marginal divergence but a clear structural shift.
>
> > Q3: Unclear scalability
>
> **CELL has the scalability potential.**
> First, the number of trainable parameters is $O(Lh^2)$, which is modest.
> Second, the main bottlenecks are cross-domain retrieval and alignment, with costs of $O((N_t^{\text{conf}})^2 h)$ and $O(N_s N_t^{\text{joint}} h)$, respectively, where $N_t^{\text{conf}}$ and $N_t^{\text{joint}}$ denote confident target nodes and jointly confident target nodes. **These operations are highly parallelizable and can be accelerated on GPUs**.
> Third, **the practical cost can be further reduced by using a smaller top-$K$ and higher pseudo-label confidence thresholds**.
>
> > Q4: Limited contribution of Lemma 3.3
>
> Eq. (24) is not a technically deep inequality, and its role is to expose the class-wise selection bias induced by confidence filtering and to connect this bias to our adaptive thresholding design.
> We will revise the presentation more precisely as: https://anonymous.4open.science/r/CELL_ICML2026-3DB5/rebuttal/lemma3-3.md
>
> > W3, Q5, Q6: Unclear proofs of Theorems 3.1 and 3.2.
>
> For Theorem 3.1, the key result is the MI-based EO bound $\Delta_{EO}\le C\sqrt{I(Z_c;S\mid Y)}$, together with the proxy assumption that (Z_s) captures (S), yielding $I(Z_c;S\mid Y)\le I(Z_c;Z_s)+\delta \le \epsilon+\delta$. We will make this assumption explicit, place the Lipschitz step correctly, and clarify the role of Eq. (32) as https://anonymous.4open.science/r/CELL_ICML2026-3DB5/rebuttal/eq32.md
>
> For Theorem 3.2, Eq. (39) is the standard DA bound in the learned representation space, while Eqs. (40)–(41) bound the additional leakage-induced residual via TV distance, Pinsker’s inequality, and Jensen’s inequality, leading to an extra $\mathcal{O}(\sqrt{I(Z_s;Z_c)})$ term. Thus, Eq. (42) is intended as a fairness-aware extension of Eq. (39), and is looser by design because it explicitly accounts for residual sensitive dependence.
> We will revise the proof to make this two-step structure explicit, distinguish the standard DA error from the leakage-induced residual term, and clarify that the final bound is intentionally looser.
>
> > Q7: Missing top-k description
>
> We revise the paper in Section 3.4:
> $K$ denotes the number of source-domain neighbors retrieved for each target node in the learned task-relevant embedding space, and the bipartite graph is then constructed by keeping only the retrieved source–target pairs that satisfy the label/sensitive constraints.
>
> > Q8: Misleading notation
>
> To avoid misleading of covariates assumption, we revise the annotations as: $P(A^{so},X^{so}) \neq P(A^{ta},X^{ta})$ and $P(Y\mid A^{so},X^{so}) = P(Y\mid A^{ta},X^{ta})$.
>
> **We hope we have addressed your concerns and would be happy to clarify any remaining points. We sincerely hope our responses support a more positive final score.**

---

> > ### Author Rebuttal · Reviewer_c8jK · 2026-04-02
> >
> > The authors have addressed my concerns.

---

> > > ### Author Response · Authors · 2026-04-02
> > >
> > > Thank you for your positive feedback and for confirming that our rebuttal addressed your concerns!
> > > We will properly include all the rebuttal concerns in the revised version, following your valuable suggestions！

---

### Decision · Program_Chairs · 2026-04-30

**Decision:**

Accept (regular)

**Comment:**

This paper studies fairness-aware graph adaptation without target sensitive attributes and proposes CELL, a causal framework combining disentanglement, pseudo-label learning, and bipartite alignment.

Reviewers generally recognized the paper’s importance, novelty, theoretical grounding, and strong empirical results. The main concerns were reproducibility, pseudo-label analysis, and experimental completeness/limitations. After the rebuttal, the authors clarified key implementation details, strengthened the pseudo-label analysis, added several recent baselines, and more clearly discussed current limitations.

Overall, I find that the rebuttal addressed most major concerns. I therefore lean toward acceptance.